# Localizing SDG 11.6.2 via Earth Observation, Modelling Applications, and Harmonised City Definitions: Policy Implications on Addressing Air Pollution

Jennifer Bailey [1,2,*], Martin Otto Paul Ramacher [3], Orestis Speyer [1], Eleni Athanasopoulou [1], Matthias Karl [3] and Evangelos Gerasopoulos [1,4]

1   Institute for Environmental Research and Sustainable Development, National Observatory of Athens, 15236 Athens, Greece
2   Herbert Wertheim School of Public Health and Human Longevity Science & Scripps Institution of Oceanography, University of California San Diego, La Jolla, CA 92037, USA
3   Chemistry Transport Modelling, Helmholtz-Zentrum Geesthacht, 21502 Geesthacht, Germany
4   Navarino Environmental Observatory, Costa Navarino, 24001 Messenia, Greece
*   Correspondence: jbailey@noa.gr

**Abstract:** While Earth observation (EO) increasingly provides a multitude of solutions to address environmental issues and sustainability from the city to global scale, their operational integration into the Sustainable Development Goals (SDG) framework is still falling behind. Within this framework, SDG Indicator 11.6.2 asks countries to report the "annual mean levels of fine particulate matter ($PM_{2.5}$) in cities (population-weighted)". The official United Nations (UN) methodology entails aggregation into a single, national level value derived from regulatory air quality monitoring networks, which are non-existent or sparse in many countries. EO, including, but not limited to remote sensing, brings forth novel monitoring methods to estimate SDG Indicator 11.6.2 alongside more traditional ones, and allows for comparability and scalability in the face of varying city definitions and monitoring capacities which impact the validity and usefulness of such an indicator. Pursuing a more harmonised global approach, the H2020 SMURBS/ERA-PLANET project provides two EO-driven approaches to deliver the indicator on a more granular level across Europe. The first approach provides both city and national values for SDG Indicator 11.6.2 through exploiting the Copernicus Atmospheric Monitoring Service reanalysis data (0.1° resolution and incorporating in situ and remote sensing data) for $PM_{2.5}$ values. The SDG Indicator 11.6.2 values are calculated using two objective city definitions—"functional urban area" and "urban centre"—that follow the UN sanctioned Degree of Urbanization concept, and then compared with official indicator values. In the second approach, a high-resolution city-scale chemical transport model ingests satellite-derived data and calculates SDG Indicator 11.6.2 at intra-urban scales. Both novel approaches to calculating SDG Indicator 11.6.2 using EO enable exploration of air pollution hotspots that drive the indicator as well as actual population exposure within cities, which can influence funding allocation and intervention implementation. The approaches are introduced, and their results frame a discussion around interesting policy implications, all with the aim to help move the dial beyond solely reporting on SDGs to designing the pathways to achieve the overarching targets.

**Keywords:** sustainable development; earth observation; air pollution; particulate matter; SDG indicator 11.6.2; urban policy; urban-scale air quality model; smart cities

## 1. Introduction: Policy State-of-Play

The discussion regarding air pollution remains as relevant and timely as it was back in 2006 when the World Health Organization (WHO) published its health-based air quality guidelines, i.e., the WHO global air quality guidelines (AQGs). The WHO only recently released updated guidelines in response to the real and continued threat of air pollution

to public health (in late September of 2021) [1]. This seminal document was produced to account for the evidence accumulated during this 15-year period on the adverse health effects of air pollution, both due to exposure to high levels of air pollution in low- and middle-income countries and reported adverse effects in high-income countries, with relatively clean air, at much lower levels than had previously been studied. These effects include seven million premature deaths every year globally (due to both outdoor and household air pollution) and millions more people falling ill from breathing polluted air [1].

Immediately prior to the AQGs' release, the European Environment Agency (EEA) published its air quality (AQ) status 2021 briefing, which will be part of the annual Air quality in Europe 2021 report [2]. This briefing refers to the older AQGs of 2006 as well as to the European Union's (EU) more lenient and politically agreed-upon Ambient Air Quality Directives and the corresponding maximum values (limit and target) that take into account technical and economic realities for their attainment across EU member states [3]. It is noted therein that although improvements have taken place, air pollution is still a major health concern for Europeans. Indicative of the differences between the WHO and EU approaches, during 2018–2019, the percentage of the EU urban population exposed to fine particulate matter ($PM_{2.5}$) was only 1–4% according to EU standards, and 61–74% according to WHO 2006 guidelines. At the same time, despite these policy-driven differences, the EEA estimated that, in 2018, approximately 379,000 premature deaths were attributable to $PM_{2.5}$ in the 27 EU Member States and the United Kingdom [4]. It is important to note that a process of revising the EU rules is underway (commenced in September 2021), initiated by the European Green Deal's umbrella of the Zero Pollution ambition, which, among others, calls for a closer alignment with the new AQGs of the WHO [5]. While the Zero Pollution Action Plan target is expected to be met if the rate (33% for the 2005–2019 period) of reducing premature deaths is to be maintained, the potential impact of observing the new AQGs for $PM_{2.5}$ would have more than doubled this rate (72%), truly serving the aforementioned aspirations [6].

Against this backdrop and partly in response to the older WHO AQGs, under the United Nations (UN) Sustainable Development Goals (SDGs) 2030 Agenda, SDG Indicator 11.6.2 (SDG 11.6.2), "Annual mean levels of fine particulate matter (e.g., $PM_{2.5}$ and $PM_{10}$) in cities (population weighted)" was designed to address the public health threat posed by air pollution, the only SDG indicator to do so. SDG 11.6.2 is a Tier 1 indicator, meaning it has a conceptually clear, established methodology, standards are available, and data are regularly produced by countries, with the WHO serving as the custodian agency (while reporting entities include ministries of environment, environmental agencies as well as national monitoring networks). In particular, countries with AQ monitoring networks provide the annual mean concentrations and corresponding number of inhabitants to derive the national population-weighted exposure to PM in cities using a generalized formula. Additional data, such as satellite retrievals of aerosol optical depth, chemical transport models, topography, etc., can be utilized in the absence of ground measurements [7]. The powerhouse behind this integration of data that ultimately delivers yearly air quality profiles for individual countries, regions, and globally, is the data integration model for air quality (DIMAQ) [8].

Despite the fact that SDG 11.6.2 is regularly produced at the country level, the UN Decade of Action is underway, during which national as well as local governments must accelerate their efforts to deliver on the SDGs, i.e., go beyond monitoring to actually lowering the indicator. Echoing this need, policy has recently refocused attention on localizing SDG reporting, without undermining the vast communication and comparability capacity the SDGs unambiguously hold. This is exemplified by the reports produced by the Sustainable Development Solutions Network (SDSN) that feature an ever-growing disaggregation, commencing from regional or national and arriving at subnational as well as city-level [9]. This emphasis on cities was famously encapsulated in UN Secretary-General Ban Ki-moon's 2012 remark "Our Struggle for Global Sustainability Will Be Won or Lost in Cities" [10]. The case for pursuing city-reporting is strongly advocated for by the SDSN, which underlines the require-

ment of engagement by local and regional governments that will ultimately implement local transformations, even more so in Europe, where these governments possess significant policy and investment levers and urban areas are home to over 2/3 of the EU's population, generate up to 85% of EU GDP, and account for about 60–80% of energy use [11,12]. Implementing the localization of SDG monitoring, SDSN has recently published the 2019 SDG Index and Dashboards Report for European Cities (prototype version) which provides an overview of the performance of 45 capital cities and large metropolitan areas [11]. The European Commission, through the Joint Research Centre, has produced the first *European Handbook for SDG Voluntary Local Reviews* to serve as a global reference for cities to monitor the achievement of the SDGs at the local scale [13]. Moving beyond SDG monitoring, the global UN initiative "United for Smart Sustainable Cities" (U4SSC) has further systematized city-level reporting by offering the international standard of the "Key Performance Indicators for Smart Sustainable Cities" [14]. These 91 KPIs, each linked to particular SDGs, aim at providing cities with a standardized method to collect necessary data and measure performance and progress as regards achieving the SDGs.

Localization of SDGs comes with unique challenges and caveats. The aforementioned SDSN report mentions constraints driven by availability, quality, and comparability of data (a problem already recognized in the Tier nomenclature of the SDGs). The smart city concept, through revolutionizing data gathering and overall endorsement of open and inclusive data practices, may alleviate part of this challenge [15]. It has recently been argued that Earth observation (EO, namely, remotely sensed and/or in situ and/or modelled data) can also effectively provide support in this direction, especially regarding geographical disaggregation of data, sustainability of monitoring workflows, and credibility of information provided [16]. The EO Toolkit for Sustainable Cities and Human Settlements further exemplifies this through several worldwide city-use cases that provide city stakeholders with curated showcases highlighting the added value that EO brings in everyday and long-term city operations, while also linking with state-of-the-art tools and specific datasets, all ready for use [17]. Apart from the above data aspects, the Leave No One Behind principle of the SDG Framework, which aims at reducing inequalities and vulnerabilities, reverberates in all EU policies, including the European Green Deal, and creates particular demands related to localization efforts at the intra-urban scale. At the same time, regarding environmental justice, a growing body of research has shown that most cities worldwide leave lower-income communities with higher shares of environmental burdens and lower shares of environmental benefits, and have proposed specific metrics to quantify this inequity, such as the Urban Environment and Social Inclusion Index (UESI) [18,19]. These metrics, however, also necessitate data in order to quantify distributional equity across neighbourhoods. Lastly, the degree to which localization of SDGs promotes inclusive development and balances a techno-managerial framework that is useful for decision makers, while also allowing for local variation, ultimately depends on the monitoring system's ability to truly capture inequalities at below-national level (intra-region or intra-urban) [20].

The above technical and inherent issues related to SDG localization are also present in the case of SDG 11.6.2, due to the nature of air pollution (such as the wide variety of emission sources and influencing factors) and PM in particular (such as differentiating the origin of air pollution in the city between urban, regional, national, and transboundary). As SDG 11.6.2 comprises both the pollution itself and the population, its study inevitably encompasses both variables and their dependence on intra-urban gradients. The literature has acknowledged this nexus at work. Wilson et al. (2005) cautions that central monitoring sites may not accurately capture spatial complexities and may ultimately lead to erroneous conclusions with respect to homogeneity of intra-urban particulate concentrations with an indirect impact on epidemiological studies [21]. Regarding the Leave No One Behind aspect, when examining the spatial variability of exposure in relation to socioeconomic indicators in Europe's metropolitan areas, Samoli et al. (2019) offered key insights of this nexus as higher ($NO_2$) pollutant concentrations were observed in areas with higher population density, and deprived European urban areas presented the lowest air quality

levels [22]. The "Urban PM$_{2.5}$ Atlas: Air Quality in European cities" report of the Joint Research Centre clearly states that determining the geographic (but also sectorial) scale of action is key to effectively address air pollution problems and ultimately, planning and impacts on air pollution, and thus SDG 11.6.2, is city-specific [23].

This study aspires to further the discussion of SDG 11.6.2, in the context of the new WHO global AQGs and the SDG Decade of Action that is underway. Triggered within the SMURBS/ERA-PLANET project, which has revisited the smart city concept via leveraging state-of-the-art EO towards methods, tools, products, and services that enhance environmental and societal resilience to the specific urban pressures of air pollution, disasters, and urban growth [16]. Here, while very sensitive to the specific challenges of urban localization, the SDG monitoring potential of EO is pursued via two modelling applications, one at the city-level and one at the intra-urban scale, by utilizing openly available EU-wide Copernicus data, heavily supported by remote sensing (RS). Moreover, as the recent study from the architect of the official UN SDG 11.6.2, who also utilized CAMS (albeit a global product) data, exemplified, the applications are replicable once certain data criteria are met, i.e., population and air pollution concentration fields—which are also scalable—when newer versions of these arise (such as finer resolution of population or concentration estimates) [24]. Lastly, the study will explore the sensitivity of the SDG 11.6.2 to the city definition and, through an intuitive portal and statistical analysis, will offer some policy insights.

## 2. Methodology

The structure of this study is as follows: we begin by laying out the methodology for calculating the SDG 11.6.2 for every city and country in Europe (Figure 1), using openly available EO data and two different city definitions, which ultimately feeds the SMURBS platform (Figure 2). This is followed by introducing two approaches for intra-urban SDG 11.6.2 calculations (Figure 3) using city scale air quality modelling in the case study of Hamburg. After illustrating the results of all approaches, the results are compared to established and otherwise published SDG 11.6.2 values, and relevant policy implications of this study are discussed.

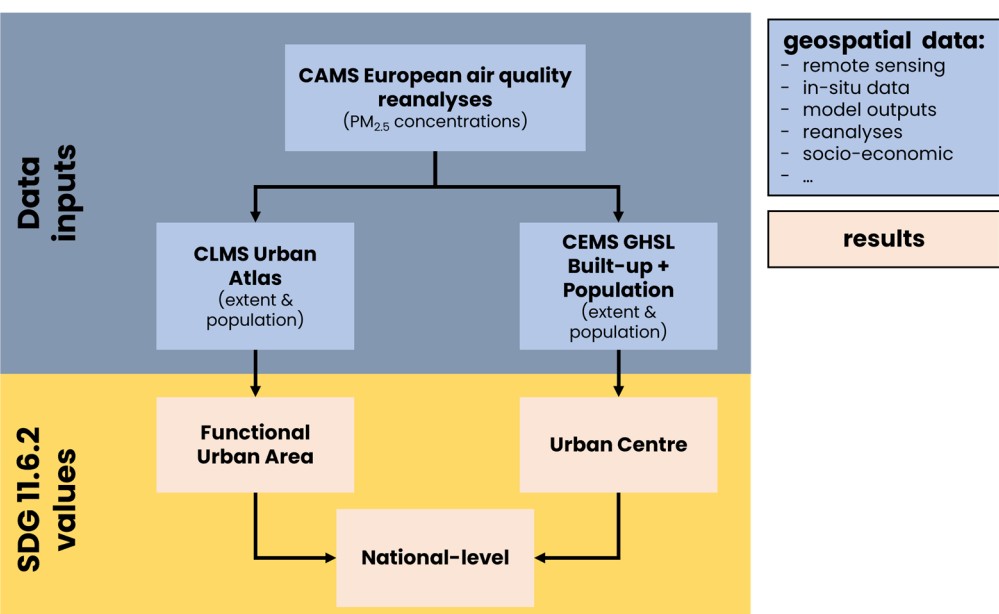

**Figure 1.** Workflow chart depicting the approach and calculation of SDG 11.6.2 values at the city and national levels using city definitions of urban centre and functional urban area, eventually incorporated into the SMURBS platform.

## 2.1. Defining SDG Indicator 11.6.2 and City Definitions

By definition, SDG 11.6.2 necessitates that a city's extent and the enclosed population are clearly defined. The official UN calculation of the national value of the indicator requires annual average concentrations of $PM_{2.5}$ as input, thus implying a certain geographical extent within which this concentration is estimated and a corresponding population:

$$Annual\ mean\ level = \frac{(C_{U1}P_{U1} + C_{U2}P_{U2} + \cdots + C_{Un}P_{Un})}{P_{total}}, \tag{1}$$

where $C_U$ is the $PM_{2.5}$ concentration of an urban area (i.e., city), $P_U$ is the population of this urban area, $n$ is the total number of urban areas in a country, and $P_{total}$ corresponds to the total population of all cities in a country [7].

While the indicator equation is defined, each country might have their own definition of an urban area. This definition could be based on a singular population threshold, population density, administrative delimitation, or some combination of criteria, which makes aggregation beyond the country level and comparison across countries ineffectual. Beyond divergence in defining urban areas, there is further discrepancy between the conceptual understanding of urban areas, which define the urban extent, producing different city boundaries and dictating how they are measured. In line with the locality of indicators within SDG Goal 11 and the aforementioned discrepancies, the UN Statistical Commission, along with others, has acknowledged the need to deviate from traditional definitions of and data collection for cities, and endorsed a harmonised spatial-based approach explicitly to facilitate calculations and comparisons such as Indicator 11.6.2 [25,26]. The degree of urbanisation (DEGURBA) methodology was put to the test, and ultimately endorsed by the UN Statistical Commission, to aid in international and regional statistical comparison in relation to city, urban, and rural delineations. DEGURBA is a classification that indicates the character of an area built upon a basis of uniform 1 $km^2$ population grid cells, classifying LAU2 (previously the lower local administrative unit consisting of municipalities) or communes into three types of areas: cities, towns and semi-dense areas, and rural areas [27].

The European Commission endorsed two objective city definitions that are extensions of the DEGURBA methodology that are utilized in this study—urban centre (UC) and functional urban area (FUA). The EC Joint Research Centre's (JRC) Global Human Settlement Layer (GHSL) extended DEGURBA by recalculating classifications using both population density criteria and density of built-up area derived from primary databases (not LAU data), producing the open and freely available Urban Centre Database (UCDB), which is in essence agnostic to national definitions [28]. The FUA exploits the UCDB, overlaying it against the LAUs, and categorizes the latter based on criteria that labels the units as cities or not (whether the type of grid cells in which the majority of LAU population resides are UC grid cells or not). A functional urban area is then defined by incorporating the commuting zone into that city's overall extent based on commuting patterns from Eurostat or country data [29].

A principal motivation of the quest for a harmonised city definition was to address the absence of such an approach in the face of several global policy agendas, such as the Global Monitoring Framework of the 2030 Agenda for Sustainable Development and the Action Framework of the Implementation of the New Urban Agenda, among others, which call for the collection and tracking of localised data [30,31]. In relation to SDG 11.6.2, the UN approach defines cities as reported per country, which influence the resulting total extent, population, and PM measurements utilized in each city's average. However, the final reported UN indicator value is one national aggregate estimate using PM values from air quality monitoring networks and, where networks are not available, using additional data, as noted in Shaddick et al., 2021 [24]. Introducing a third city definition with relation to SDG 11.6.2, Eurostat, charged with monitoring EU progress towards sustainable development and having adopted their own EU SDG indicator set (in the European-level policy frame), monitors an analogous indicator denoted as sdg_11_50 [32]. The Eurostat workflow entails annual mean concentrations of PM at urban background stations from the

EEA AQ e-reporting database that are located within the so-called agglomerations as their city definition. An agglomeration is defined as a conurbation with a population larger than 250,000 or with a smaller population with a given population density per km$^2$ established by the EU member states [33].

### 2.2. SMURBS Estimation of SDG 11.6.2 at the Country and City Level across Europe

Under the SMURBS/ERA-PLANET project that advocated for the role of EO in the smart city paradigm, an EO-driven methodology was created, based on the previously mentioned EC-endorsed city definitions, for acquiring the value for 11.6.2 across Europe at country and city level [16,34]. Figure 1 displays the workflow eventually producing city and country level SDG 11.6.2 values across Europe, which feeds the SMURBS SDG Indictor 11.6.2 Earth Observation Platform (Figure 2). We carry this out through utilizing services built upon remote sensing and other EO data described below to calculate the indicator and display the values on an online platform.

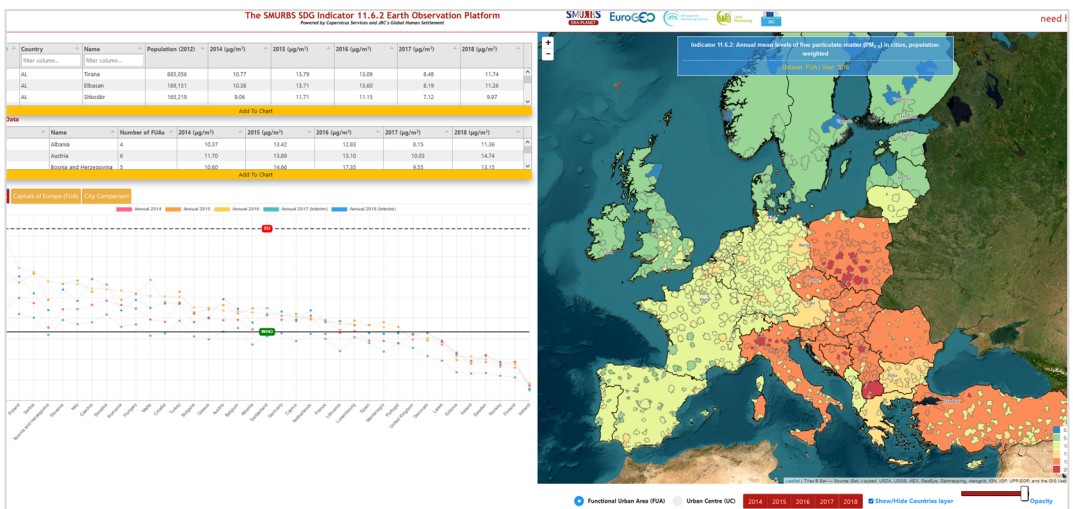

**Figure 2.** Interface of the SMURBS SDG Indicator 11.6.2 Earth Observation Platform [34].

### 2.2.1. Copernicus Services, Data Provision, and Processing

Consistent, harmonised, and operational data are prerequisites in the effort to localise the SDGs. The Copernicus Programme, one of the main contributions of the EU in the rapidly expanding domain of EO, and its driving initiatives, such as the Group on Earth Observations, offers such data, accompanied by a free and open policy [35]. Within the SMURBS platform, the population and city outline data pertaining to the UC definition are provided by the JRC's Global Human Settlement Layer (GHSL) and UCDB, which incorporate human settlement information from satellite imagery and modelling, and fine scale population data from censuses, downscaled to a uniform grid of 1 km$^2$, with a reference year of 2015 [28]. The GHSL is part of the Copernicus Emergency Management Service (CEMS). The FUA definition area and population data are provided by the Copernicus Land Monitoring Service (CLMS) Urban Atlas, with 2012 as a reference year [36]. All Copernicus services and, therefore, the data products utilized in this study, combine in situ measurements with remote sensing imagery performed by Sentinel satellites, as this is necessary to validate the models and calibrate the satellite sensors.

Regarding the required PM$_{2.5}$ concentrations, the platform for both approaches exploit air quality data from the Copernicus Atmosphere Monitoring Service (CAMS) regional ensemble reanalysis product, which offers concentrations throughout Europe at a 0.1° × 0.1° (or 11 km) spatial resolution for 2014–2016 (validated data) and 2017–2018 (interim data) [37]. Notably, this regional product should not be confused with the global CAMS product utilized in Shaddick et al. (2021) [24]. This product perfectly exemplifies

the combined use of EO platforms as it entails modelled information based on an ensemble of chemistry-transport models (CTMs) integrated with in situ or satellite information.

The Copernicus data were then processed via standard GIS tools to ultimately create a relational database management system (RDBMS) that feeds the platform via a map server middleware, both for the UCDB and FUA approaches. The downloaded NetCDF files from CAMS, containing the air quality information for each year, were first transformed to raster images, which were then converted to the WGS84 coordinate system. Each image was vectorized and the grid values were incorporated into an attribute table. Each cell within the UC/FUA shapefiles (also containing population information, i.e., the $P_i$) was analysed to calculate the surface ratio between sea and land. Based on these grid cells and utilizing area averaged weights for the cells at the border, basic statistics were produced, i.e., the concentration averages (the $C_i$ discussed above) and minimum/maximum values. The calculation algorithm was employed for each country for both approaches. For each shapefile, a product of concentration multiplied by the shapefile's population was calculated, then added with the rest of the products (the numerator of Equation (1)). Results were stored in separate files along with accompanying information (such as the number of UC/FUAs that went into the calculation of the national value but and the value per city—in essence the annual concentration average).

Based on the city definition (UC or FUA) and year of interest (2014–2018), the population-weighted average annual particulate matter concentrations in cities and countries are displayed in the SMURBS online platform via a collection of visualization tools. Data tables display indicator values for every European city (more than 800 UCs and 750 FUAs) and country (37 in total), linked to an interactive map delineating each city's boundaries and assigning each country a colour based on the country's indicator performance. An overview chart provides country indicator values over time, allowing for specific annual selections and comparisons across cities. On every chart, indicator values are viewed in line with the EU standard and the 2006 WHO AQGs (the novel WHO AQGs were not yet published) for $PM_{2.5}$, easily placing performance within the context of relevant policy frames. A screen capture of the platform can be seen in Figure 2. The overall scope of the platform was to present an elegant tool for calculating SDG 11.6.2, demonstrating its sensitivity to the city definition, and allowing a comprehensive policy view to the inner workings of the indicator and the potential of EO to demystify these. The values visualized in the web platform are first stored in a geodatabase and are utilized for all comparisons in the manuscript.

### 2.2.2. Qualitative Comparisons with Data from Existing Efforts

One of the main aims of this study is to compare the SMURBS-derived indicator results with the existing official UN workflow. The UN-reported 11.6.2 Indicator values come from the WHO-maintained air quality database, where designated data providers submit their indicator-related data every two years, which include ground measurements from monitoring networks supplemented by web searches [38]. For this study, we utilize the UN indicator value that pertains to urban areas within a country, as opposed to the all-area UN value, which is also published. The other country-analogue of interest for this comparison is Eurostat's sdg_11_50 indicator, which employs in situ measurements of $PM_{2.5}$ concentrations from the EEA's AQ e-reporting database, but in this case, the Eurostat-defined agglomerations are used as the urban definition. Comparisons were only implemented for the year 2016 where all approaches have produced values either officially or using validated data (extending upon what was carried out in Bailey et al., 2020) and can be seen in Section 3.1 [39].

Addressing the need for localizing the SDGs and zeroing in at the city level, we compared the SMURBS-produced values according to both urban centre and functional urban area definitions. The choice of cities to include in this study's comparison was based on the cities used in a similar effort by the SDSN in their SDG Index for European Cities, which comprises 45 European Cities (Figure A1) [11]. The SDG Index for European Cities utilizes $PM_{2.5}$ (or $PM_{10}$ converted to $PM_{2.5}$ using national conversion factors) data

published in the 2016 version of the WHO air quality database (which includes data for some cities for the year 2013), wherein the data mostly originates from the EEA, with varying city definition dependent on reporting (i.e., Eurostat's NUTS or agglomeration). While this information is not directly comparable with the platform's UC and FUA values, due to both temporal limitations and differing city definitions, we instead depict the SMURBS values in (Section 3.2), using the same 45 European cities from the SDSN Index to complement their effort with respect to its policy usefulness and implications. In line with this, we examined the distribution of differences in PM$_{2.5}$ between the SMURBS UC- and FUA-derived SDG 11.6.2 values averaged for 2014–2016 for the same selection of cities to assess the similarities/dissimilarities of the yields from each approach (Section 3.2).

### 2.3. Intra-Urban Scale Adaptations of SDG Indicator 11.6.2

The interface between the platform presented above and this second, modelling-enabled approach, is the city. Both approaches deliver SDG 11.6.2 estimates for a particular city, i.e., a single value. Considering the intra-urban scale, the calculation of urban weighted annual mean levels of fine particulate matter within one city is necessary for localised measures to reduce levels of air pollution and spearhead the Decade of Action by municipalities and relevant actors, as discussed in the introduction. The following subsections provide an overview of city-scale air quality modelling (Section 2.2.1) employed to adapt 11.6.2 to different urban scales (Section 2.2.2), and the workflow of the intra-urban approaches is illustrated in Figure 3.

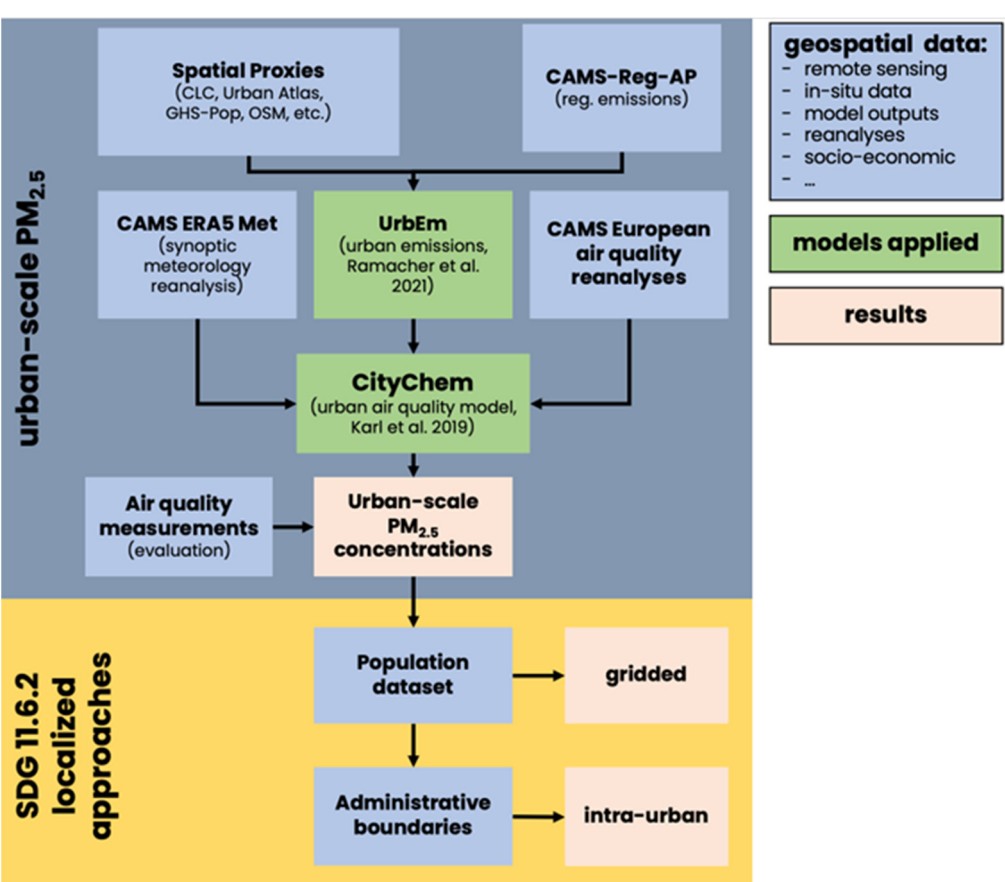

**Figure 3.** Workflow chart of the introduced intra-urban approaches to localise the SDG 11.6.2.

### 2.3.1. City-Scale Air Quality Modelling

To pursue and achieve targets of annual mean levels of PM within one city, concentration values at a city-scale resolution are necessary. Such information can be obtained by

applying city-scale air quality models, such as the EPISODE-CityChem model applied in this study to acquire the necessary concentration fields [40].

The EPISODE model with the CityChem extension is a chemistry-transport model designed especially for capturing city-scale dispersion and photochemistry of pollutants and is particularly useful for studies of aerosols in urban areas [40,41]. EPISODE-CityChem was created to fill the gap between regional and micro-scale simulations for air quality and its output includes hourly or time-averaged concentrations of pollutants at specified locations within an urban area. The performance of EPISODE-CityChem and its compliance with quality criteria for regulatory purposes has been demonstrated in several urban-scale studies [40,42–47].

The application at hand utilizes EPISODE-CityChem to simulate $PM_{2.5}$ concentrations for a $30 \times 30$ km$^2$ domain covering Hamburg (Germany) with a spatial resolution of 100 m, which serves as an example city for this study. Hamburg was chosen as a case study because the high-resolution mapping of air pollution and SDG 11.6.2 is a demanding task (as discussed in Section 4.1), which necessitates a pre-establishment of the utilized methods for any urban area of study. The proposed modelling system for the intra-urban approaches is state-of-the-art and is already applied, tested, and demonstrated for the city of Hamburg [40,42–47]. The overarching scope of this paper is to provide a demonstration of the EO potential in addressing and localizing the SDGs and, in particular, SDG Indicator 11.6.2. By using the well-established example of Hamburg, it is not necessary to demonstrate the modelling of $PM_{2.5}$ for Hamburg specifically; instead, focus can be directed towards the introduced approach for localizing SDG 11.6.2, which is designed to be generally applicable to other cities in Europe.

As was the case for the platform presented above, the EPISODE-CityChem setup incorporates Copernicus data (which are mostly based on, and include, RS data) from different Copernicus services (Figure 3). To account for regional boundary concentrations for every hour of simulation, the CAMS regional air quality (regional ensemble) is interpolated to the horizontal and vertical resolution of the domain, to be considered at the lateral and vertical borders of the Hamburg domain. It should be noted that the CAMS reanalysis predictions used in the current study assimilate measurements carried out in situ with the remote sensing imagery performed by satellites. Through the combination of multiple EO platforms, CAMS is able to deliver reliable information in real time, the future, or the past.

The urban scale emission inventory is based on the European emission inventory CAMS-REG-AP [48], which is downscaled to the urban scale [45] under application of spatial proxies, such as Copernicus Corine Land Cover (a database produced by the visual interpretation of high-resolution satellite imagery) and Urban Atlas [36], as well as GHSL population data (imagery-derived datasets) from CEMS [26]. Finally, meteorological fields to run EPISODE-CityChem simulations are derived from ECMWF ERA5 synoptic-scale reanalysis distributed by C3S [49] under application of the meteorological module of the TAPM model [50]. An overview of the CityChem model setup can be found in Appendix A (Table A1).

The simulated hourly results for $PM_{2.5}$ in Hamburg for the year 2016 were compared to air quality data from the local air quality network at five monitoring stations, measuring $PM_{2.5}$ (daily mean). The main features of model evaluation can be found in Appendix A (Appendix A.2) [44]. At all measurement sites, modelled vs. measured $PM_{2.5}$ values show FAC2 (fraction of modelled values within a factor of 2 of the observed values) values of 0.78–0.88 for daily $PM_{2.5}$, which satisfies the urban-scale air quality modelling acceptance criteria of FAC2 > 0.3 for urban dispersion model evaluation [51]. For an in-depth evaluation of the modelled concentrations for Hamburg in 2016, we refer to Ramacher et al. 2020 [44]. The simulated $PM_{2.5}$ concentration field is used to estimate 11.6.2 for the year 2016 under different intra-urban approaches.

2.3.2. Intra-Urban Scale via the Administrative and Gridded Approaches

As mentioned, there is a need for localised approaches to the SDG Indicator 11.6.2. Therefore, two intra-urban scale approaches and their application to the example city of Hamburg are demonstrated below.

To calculate SDG Indicator 11.6.2 within the city of Hamburg and as a single value for the whole city, two spatial approaches are employed: (1) an administratively oriented approach that leads to population-weighted $PM_{2.5}$ concentrations per district, neighbourhood, or other statistical unit; and (2) a gridded approach, which calculates population-weighted $PM_{2.5}$ concentrations per simulation grid cell. Both approaches utilize the calculation algorithm (Equation (1)), where mean concentration (C) and population sum (P) refer to the geographical unit at hand or the grid cells, respectively. As a general approach to match the population for different geographical units with urban scale air quality modelling results, the population is projected to the urban domain's projection, cropped to the extent of the urban domain, and, finally, resampled bilinearly to the urban domains resolution.

For (1), the administratively oriented approach, available administrative information alongside population counts are applied, as demonstrated for the city of Hamburg [52,53] for districts (Figure 4a) and neighbourhoods (Figure 4b).

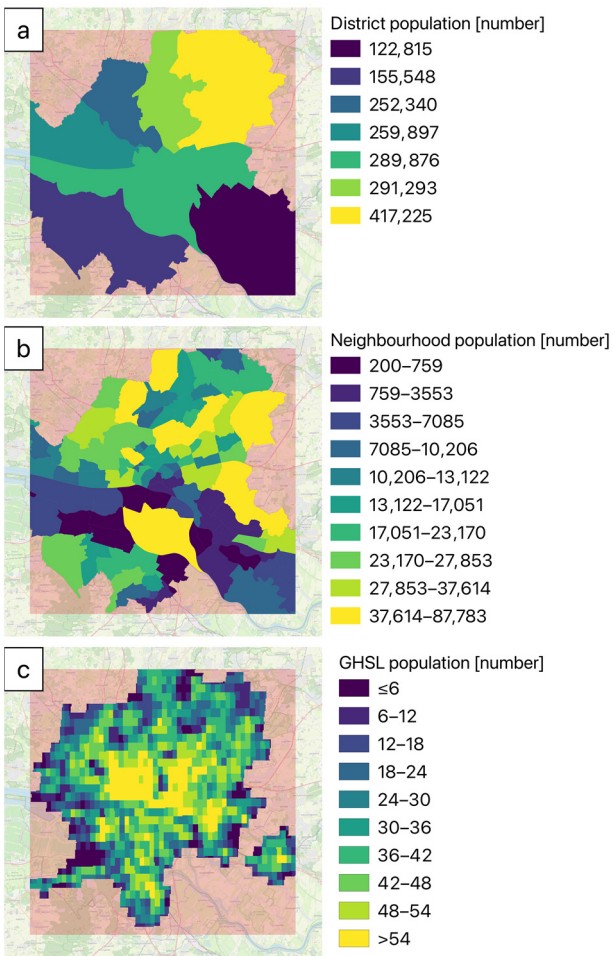

**Figure 4.** Population in Hamburg as derived from the census in Hamburg districts (**a**) and neighbourhoods (**b**), as well as population density of the Global Human Settlement Layer (GHS-POP) intersected with the Urban Centre Database (GHSL UCDB) spatial definition of urban centers (**c**). (**a**–**c**) are resampled to the urban domain of Hamburg (red transparent background) as applied in the EPISODE-CityChem simulations (30,000 × 30,000 m² extent, 100 × 100 m² resolution).

The gridded approach (2) is based on the GHSL UC definition and population dataset and, therefore, not limited to the availability of datasets on administrative boundaries [28]. Thus, the gridded approach utilizes information with a consistent grid definition, which is publicly available and can be applied to any urban area. Figure 4c shows the GHSL population of the urban center of Hamburg, embedded in the simulation domain of the EPISODE-CityChem model.

Both approaches can be applied to any city of interest, allowing for the estimation of a single 11.6.2 for the entire city, as well as an intra-urban mean $PM_{2.5}$ concentration for different spatial units, thus enabling the delivery of an indicator tailored to the jurisdiction of any competent decision-making body that implements pollution mitigation measures and urban planning.

## 3. Results

### 3.1. Country-Level Comparisons

Beginning with the country-level, Figure 5 depicts all approaches (SMURBS FUA and UC, Eurostat, and UN) for calculating the national average for SDG 11.6.2 plotted against the WHO guideline and the EU limit. WHO interim targets (ITs), which have been defined and set to support and push forward incremental milestones of lower PM values, are also displayed. To compare the outcome values of these differing approaches, it is worthwhile to determine the extent to which the values differ within a country and between countries, and, further, if these differences result in an outcome with direct policy implications, such as the country's value being above or below the EU or WHO limits, depending on the approach used to calculate the "same" indicator. In this regard, it is also essential to obtain an idea of how the Eurostat and UN values compare with each other. The overall trend for the 28 countries, with both UN and Eurostat values, is that Eurostat is, on average, 0.5 $\mu g/m^3$ (or 2.5%) higher than the UN SDG 11.6.2, while for 50% of the countries (i.e., the interquartile range), the difference (Eurostat minus UN) lies between –0.7 and 1.6 $\mu g/m^3$.

Based on the inspection of the differences between the UN and Eurostat values (as well as FUA vs. UC values) a threshold of 2 $\mu g/m^3$ (and 15% in relative difference) was selected. This threshold was based on the following semi-empirical criteria: (1) the level of 2 $\mu g/m^3$ (and, similarly, the 15% relative difference) was found to frame cases which differed markedly from the majority of pairs; (2) these high values corresponded to approximately 20% of the whole (80% percentile), which left adequate numbers of cases to draw conclusions and discern patterns from; (3) given the lower detection limit of common reference $PM_{2.5}$ analysers (order of 1 $\mu g/m^3$) and the lower threshold in terms of the WHO (5 $\mu g/m^3$). Based on the above and given that similar distributions of differences were found in all types of comparisons, we adopted the rounded values of 2 $\mu g/m^3$ (and 15%) for all relevant analyses for homogenization purposes.

Applying a threshold of 2 $\mu g/m^3$ in absolute value of the differences (the 80th percentile value of the distribution of differences using the UN value as reference) and a percentage difference of 15% (the 80th percentile values of the distribution of the percentage differences using the UN value as reference), 6 countries exceed both criteria, and 5 of those hold the pattern of larger Eurostat values compared to UN. Except for Luxembourg, the countries have $PM_{2.5}$ levels above the average levels for their respective indicators and fall within the range of 15–22 $\mu g/m^3$, without any obvious geographical pattern.

As expected, for the 37 countries depicted in Figure 5, UC values are, on average, 1.3 $\mu g/m^3$ (or 13%) higher than FUA values, while for 50% of the countries, the difference (UC minus FUA) lies between 0.6 and 1.8 $\mu g/m^3$. Applying the same combined criterion as above (i.e., 2 $\mu g/m^3$ in absolute value and a percentage difference of 15%), for the sake of comparability across indicators, and as they follow more or less the same distribution, there are 7 cases of exceedance. In these cases, the maximum difference is as much as 5.6 $\mu g/m^3$, whereas the majority of the countries (5 out of 7) have $PM_{2.5}$ levels well above the average (ranging between 14–30 $\mu g/m^3$). Furthermore, all of these cases are Balkan countries, including Turkey. Despite the low $PM_{2.5}$ levels (2.5–8 $\mu g/m^3$) for the remaining

2 countries in this category (Norway and Iceland), the UC values are 53% and 90% higher compared to the FUA values.

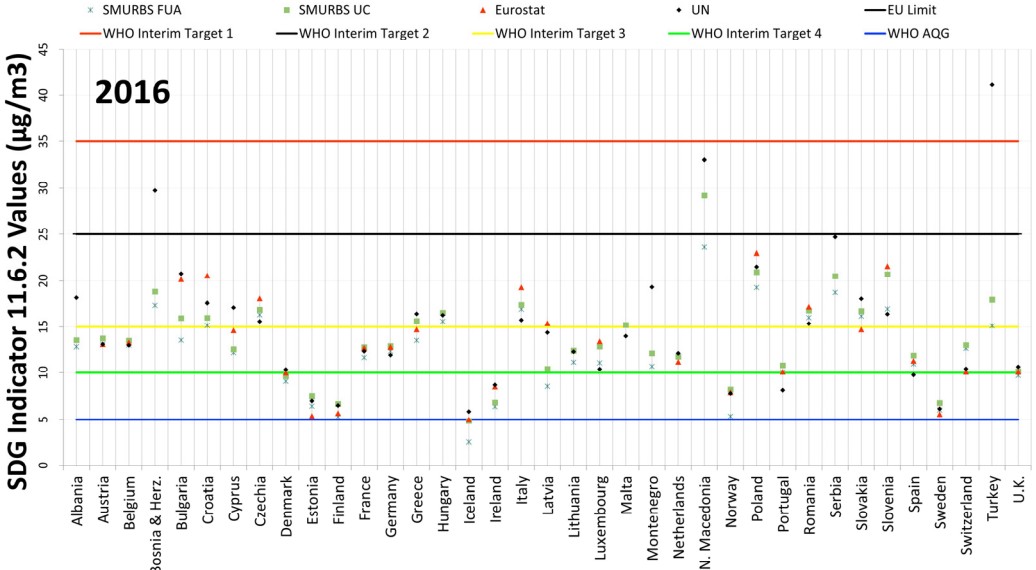

**Figure 5.** SDG Indicator 11.6.2 values estimated via the different country-level approaches (UN, Eurostat, SMURBS-FUA and UC) for 2016, per country. The EU limit and WHO AQG values for annual average PM concentrations are also shown, along with WHO interim targets (IT). The EU limit and WHO IT 2 are the same value (25 µg/m$^3$). Nine country values were not reported by Eurostat for 2016, two of which are EU-28 countries.

The values derived from the SMURBS methodology generally tend to underestimate the official Eurostat and UN values. Further, on average, the values largely follow a pattern of FUA < UC < UN < Eurostat, however, it should be noted that when comparing the UN and Eurostat values (the latter reported for only 28 countries), just over half of the time (53% of countries), the Eurostat values are larger, almost demonstrating balance. The average difference (UN minus UC) is 1.4 µg/m$^3$ (or 9%), while for 50% of the countries, the difference lies between −1.0 and 1.8 µg/m$^3$. The UC value is, on average, smaller than the UN value; nevertheless, the 2 values again exhibit a balanced distribution: the UC value is larger than the UN value for just over 50% of the countries. In total, 5 of the 13 countries that meet the combined criteria (applied previously) also show this trend of UC overestimation of the UN value and, for Switzerland only, the criterion would also be met if FUAs were used instead of UCs. For those 5 countries (Spain, Portugal, Luxembourg, Switzerland, and Slovenia), the underestimation ranges from around 2 to 5 µg/m$^3$ (18% to 25%). The employment of FUAs in the criterion would add 6 countries to the exceedances list, namely, Greece, Croatia, North Macedonia, Ireland, Norway, and Iceland. The latter three are bordered by the Atlantic Ocean and the former three are in south-eastern Europe, and such geography could further influence PM$_{2.5}$ values when alternate city definitions are used. If FUAs were utilized in the criterion, four countries would then be dropped from the exceedance list (Spain, Portugal, Luxembourg, and Slovenia). It is notable that six out of eight of the remaining thirteen countries that meet the criterion and exhibit larger UN values than the UC value are Balkan countries (only Latvia and Cyprus remain).

Alternatively, in comparing the UC values with Eurostat values where possible, the average difference (Eurostat minus UC) is 0.41 (or 2.1%) and the difference for 50% of the countries falls between −0.66 and 1.3 µg/m$^3$. While the average Eurostat value is larger in magnitude than the UC, the values are evenly distributed, meaning that exactly half of the countries have larger UC values and the other half maintain larger Eurostat values. Approximately 20% (or 6) of the countries meet the criteria and 4 of those countries, i.e., Switzerland, Cyprus, Bulgaria, and Latvia, are the same as in the UC and UN compari-

son. Although the results are limited to the 28 countries with reported Eurostat values, 2 additional countries now meet the criteria, and while in the case of Estonia the discrepancy would be resolved if FUAs were used instead (it should be noted that both the FUA and UC values are larger than the Eurostat value), Croatia would still meet the criteria using FUAs (and the Eurostat value remains larger than the FUA value). Simultaneously, in comparing the UC and Eurostat values, three countries who previously met the criteria in the UC/UN comparison are eliminated from the list (Spain, Portugal, and Luxembourg). In comparing FUA and Eurostat values, there exists a larger discrepancy (1.55 and 16.1%), for which 11 countries meet the criteria, providing justification for the focus on UC values in this comparison.

Taking the comparison a step further, we have assessed whether the selection of the different SDG 11.6.2 value sources dictates a threshold (EU, WHO, and WHO Interim Targets 1–4) exceedance. From the four values (SMURBS FUA and UC, Eurostat, and UN), we isolated the one(s) driving the conclusion (and thus, potentially driving decision-making) to be on the opposite side of a threshold compared to the rest of the values, with the arrow denoting what would be the "misleading" conclusion, or the one standing out from the rest. For example, in the case of the United Kingdom, the entry should read: "if the FUA value was used instead of the other three, then the WHO Interim Target 4 would not have been exceeded." To verify whether the higher variability between the four values is responsible for the policy implications, the standard deviation of these values as a percentage of their average is also provided in Table 1.

Policy implications were identified in 16 out of 37 countries. As the vast majority of countries are above the 5 $\mu g/m^3$ WHO threshold, no policy implications are observed around this guideline, besides the fact that countries must intensify their efforts in order to achieve the target. The only exception is Iceland, which presents quite a high variability (30.9%) between the notably low indicator values (all approaches yielding results ranging from 2.6 to 5.9 $\mu g/m^3$). In this case, only if the UN value is used, then the threshold is exceeded. Similarly, the WHO Interim Target 1 (IT1) of 35 $\mu g/m^3$ is only exceeded once, in the case of Turkey when the UN value is utilized, while both FUA and UC values classify the country just above the WHO IT3. Apart from Turkey, the only countries for which implications exist with respect to the EU threshold of 25 $\mu g/m^3$ (coinciding with the WHO IT2) are Bosnia and Herzegovina, which exceeds the threshold if the UN value is used, and North Macedonia, which would meet the target by adopting the FUA approach. In all these cases, the percentage of variability is high (16.6 to 57.8%).

The majority of implications are found around IT3 and IT4 of the WHO, as those are close to the average PM$_{2.5}$ levels of most European countries. Overall, there are three cases (Spain, Portugal, and Malta) where the UN value would lead to the conclusion that the respective target is met, and another three (Cyprus, Albania, and Montenegro) where the UN value would lead to the opposite conclusion. Interestingly, the first case is related to relatively lower variability and the countries are in the south-western part of Europe (or central west Mediterranean). The latter case presents higher variability percentages and the countries are geographically situated in the south-eastern part of Europe (the Balkans and eastern Mediterranean). More implications for these two WHO ITs (3 and 4) are related to the use of the FUA approach which, as expected, results in the accomplishment of targets when used. Finally, the dependence of implication occurrence on SDG values' variability is quite strong. Where the variability is less than 15%, a quarter of the countries had a type of implication, while where the variability is more than 15%, the percentage of countries with implications soars to 80%.

**Table 1.** Policy implications of the approaches used to calculate SDG 11.6.2 (2016), according to the "misleading" value(s) crossing thresholds (WHO AQG and interim targets (IT), EU limit), including countries that do not have reported Eurostat (EuSt) values for comparison, are denoted. The direction of the arrow indicates whether the corresponding value crosses above or below a threshold. The standard deviation of all (three or four) indicator values over the average of those values is provided, also for countries with no threshold exceedances (listed sequentially from smallest to largest per category).

| Limit, Guideline & Targets: | | WHO | WHO IT4 | WHO IT3 | EU/WHO IT2 | WHO IT1 |
|---|---|---|---|---|---|---|
| Country | SD/AVG | 5 µg/m³ | 10 µg/m³ | 15 µg/m³ | 25 µg/m³ | 35 µg/m³ |
| UK | 3.7% | | FUA ↓ | | | |
| Denmark | 5.6% | | FUA, UC ↓ | | | |
| Spain | 8.2% | | UN ↓ | | | |
| Greece | 8.3% | | | FUA, EuSt ↓ | | |
| Slovakia | 8.4% | | | EuSt ↓ | | |
| Portugal | 11.9% | | UN ↓ | | | |
| Cyprus | 16% | | | UN ↑ | | |
| Bulgaria | 19.7% | | | FUA ↓ | | |
| Latvia | 26.6% | | FUA ↓ | EuSt ↑ | | |
| Iceland | 30.9% | UN ↑ | | | | |
| Malta * | 4.7% | | | UN ↓ | | |
| North Macedonia * | 16.6% | | | | FUA ↓ | |
| Albania * | 19.6% | | | UN ↑ | | |
| Bosnia and Herzegovina * | 30.8% | | | | UN ↑ | |
| Montenegro * | 32.9% | | | UN ↑ | | |
| Turkey * | 57.8% | | | | UN ↑ | UN ↑ |
| | 0–5% | Belgium, Austria, Netherlands, Germany, France ǀ Hungary *, Malta * | | | | |
| | 5–10% | Romania, Czechia, Poland, Sweden, Italy ǀ Lithuania * | | | | |
| | 10–15% | Finland, Luxembourg, Switzerland, Estonia, Slovenia, Croatia ǀ Serbia * | | | | |
| | >15% | Ireland, Norway | | | | |

\* Non-EU28 country and, therefore, does not have a reported Eurostat value.

### 3.2. City-Level Comparisons

To further assist the localised aspect of the discussion around SDG Indicator 11.6.2, European cities matching the SDSN SDG Index cities were used and their SMURBS approach outcomes visualized and compared. Similar to the country comparison, Figure 6 shows the UC and FUA values for the indicator per city for the last validated year of 2016, barring the Hague, as it is not considered to be its own urban centre, but a part of Rotterdam, according to the UCDB. The only value just slightly above the EU limit, and, therefore, also the WHO IT2, is the UC value for Milan (25.03 µg/m³). In total, 50% of the cities have values that fall squarely between the WHO IT 3 and 4, while 20% of cities fall between IT 2 and 3. The cities that crossed target lines always were above if the UC definition was considered and below if the approach utilized the FUA definition, which is part of a larger trend, where 95% of cities values were higher if defined by its UC. While no cities' values met the WHO AQG, almost 15% of them had values between the IT4 and the AQG.

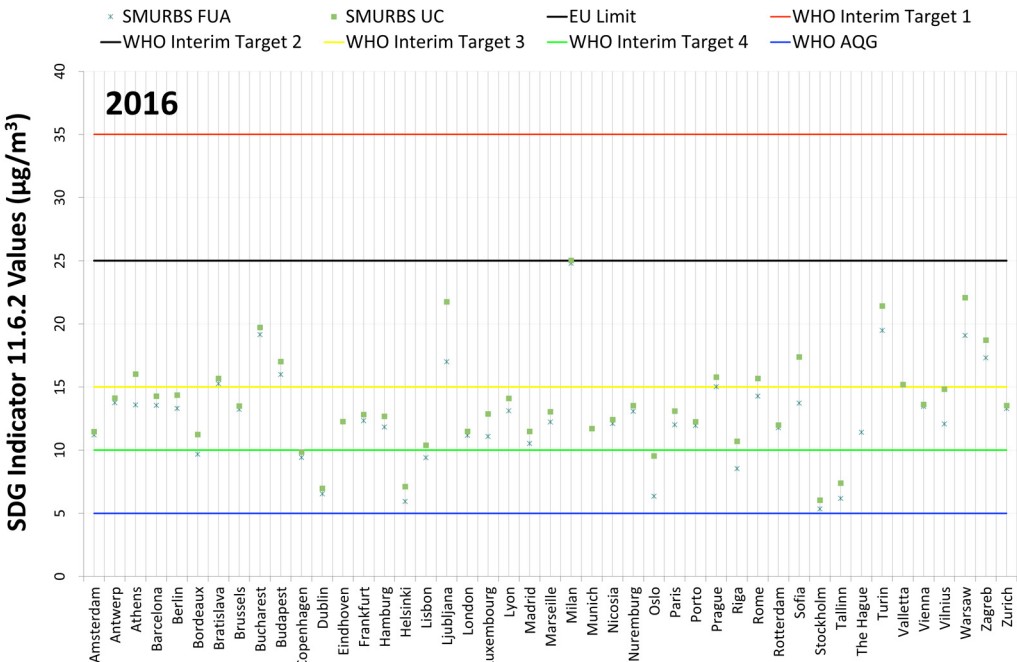

**Figure 6.** Comparison of SMURBS UC and FUA indicator values for European cities along with the EU limit, WHO AQG, and WHO interim targets.

In most cases, the two SMURBS values are very similar; however, this is not true in all cases. To discern this pattern, the distribution of differences provided in Figure 7 shows the difference between the two approaches for cities averaged across years with validated data (2014–2016). While the majority of cities maintained values for Indicator 11.6.2 that were very similar, the approaches diverged by 2–3 $\mu g/m^3$ for Warsaw, Oslo, Athens, and Riga. The largest differences were seen for Ljubljana and Sofia.

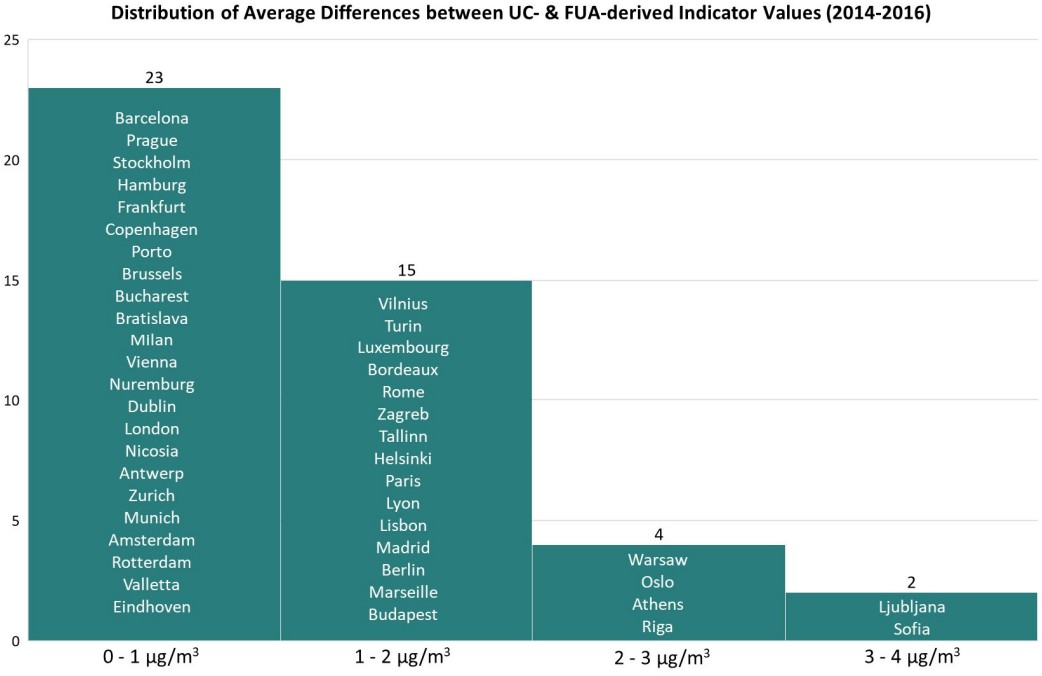

**Figure 7.** Distribution of differences in $PM_{2.5}$ between the SMURBS UC- and FUA-derived SDG 11.6.2 values for European cities, averaged for 2014–2016, listed from highest to lowest difference between the approach yields.

### 3.3. Intra-Urban Scale SDG 11.6.2

#### 3.3.1. PM$_{2.5}$ Modelling Results for Hamburg

Both intra-urban approaches to derive the SDG Indicator 11.6.2 are based on the PM$_{2.5}$ concentration fields simulated with EPISODE-CityChem. For the simulated concentrations in the city of Hamburg, the PM$_{2.5}$ concentration mapping at a resolution of $100 \times 100$ m$^2$ reveals a very heterogeneous distribution, with 2 hotspots in industrial areas, and hotspots close to the port area, near a highway intersection, and near a waste-to-energy plant (Figure 8). In addition to these hotspots, the simulations show elevated levels of PM$_{2.5}$ in the city center (annual mean of up to 15 µg/m$^3$) as well as over and adjacent to the road network (annual mean of up to 30 µg/m$^3$). Such a scale of detail is enabled only via high-resolution air quality modelling with the capacity to treat road activity in a linear format.

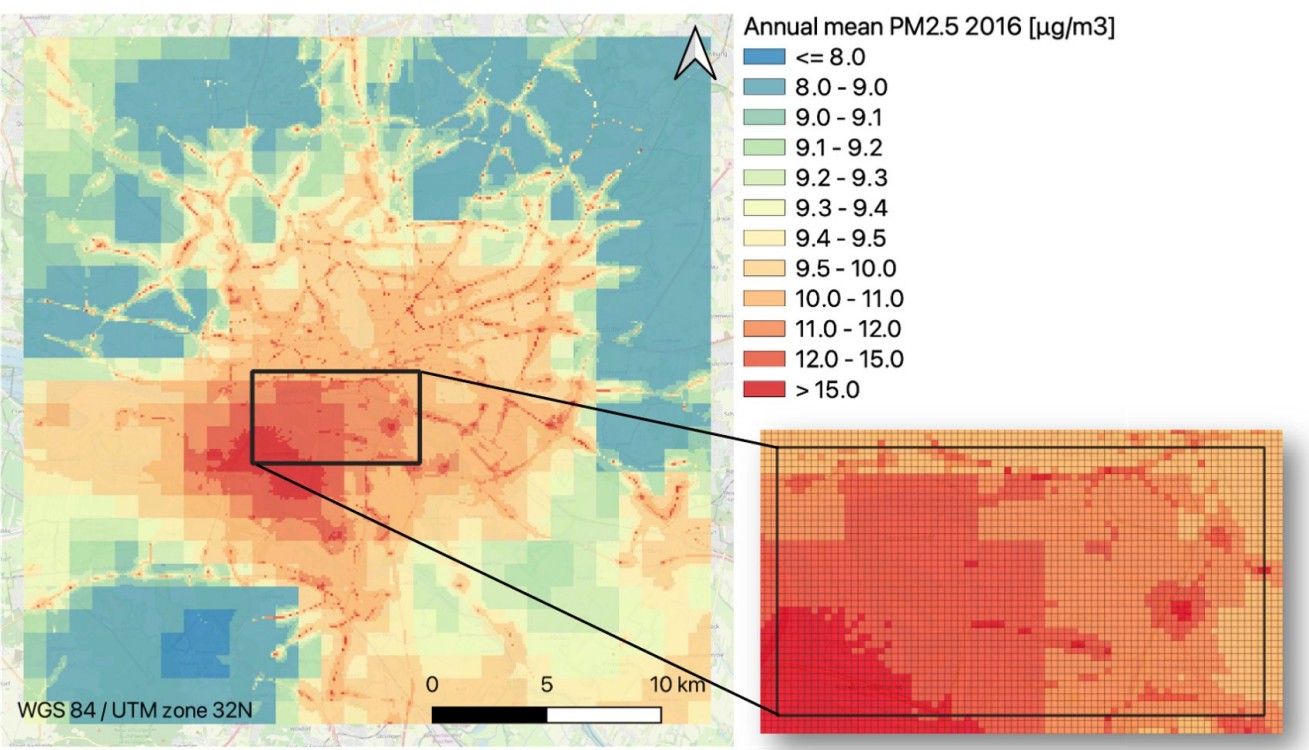

**Figure 8.** Annually averaged PM$_{2.5}$ concentrations for the $30,000 \times 30,000$ m$^2$ Hamburg domain with a resolution of $100 \times 100$ m$^2$, as simulated with the EPISODE-CityChem model.

#### 3.3.2. Administrative-Oriented Approach to SDG 11.6.2

The administrative-oriented approach was applied to PM$_{2.5}$ concentration fields for Hamburg 2016 on both district and neighbourhood units, as explained in Section 2.3.2 (Figure 9). The application of the adapted SDG 11.6.2 algorithm led to the following values of the indicator:

- 11.51 µg/m$^3$, based on district-level population and PM$_{2.5}$ means;
- 11.83 µg/m$^3$, based on neighbourhood-level population and PM$_{2.5}$ means.

As expected, values are quite close (2.3% difference); however, it is shown that the available spatial resolution and/or the selection for aggregation of the input parameters (population density and PM$_{2.5}$ concentrations) can influence the indicator's value. This reveals the current need for in-depth spatial analysis of PM$_{2.5}$ exposure within cities' boundaries, which can be performed by analysing the annual mean PM$_{2.5}$ concentrations per district and neighbourhood (Figure 9).

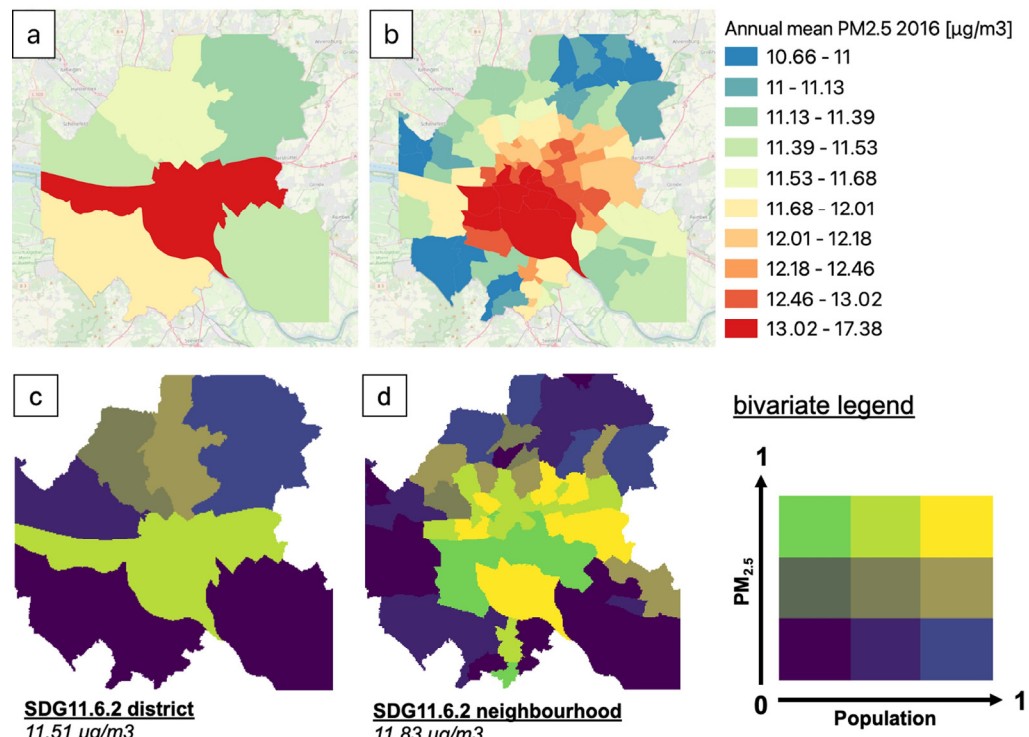

**Figure 9.** Annual mean PM$_{2.5}$ concentrations in districts (**a**) and neighbourhoods (**b**) for Hamburg, 2016, as simulated with EPISODE-CityChem. Bivariate plots for normalized mean PM$_{2.5}$ concentrations and normalized population density in districts (**c**) and neighbourhoods (**d**) for Hamburg, 2016. The SDG 11.6.2 value for the urban area, deriving from district and neighbourhood aggregations and calculations provided at the bottom.

Additionally, by studying Figure 9, it becomes evident that increasing the number of urban zones leads to a more detailed insight of the spatial variability of people's exposure to pollution, which is a valuable piece of information for targeted mitigation or other policy measures required for efficient air quality plans, especially in dense areas with elevated pollution. For example, it becomes evident that in Hamburg, the neighbourhood with the highest PM$_{2.5}$ concentration is at the same time the one with the lowest population, thus enabling the identification of the real hotspots of human exposure (yellow areas on the map) and the design of holistic approaches to protect the people where they live (or work) and cut emissions where the sources are.

Although it is often an advantage in urban planning and policy to have results based on a district or neighbourhood level, especially when these units can act as standalone decision-making centres, it is equally important to understand the extent to which an even higher resolution, in support of the aggregated calculations, would really add value to delineating human exposure aspects or unlocking perspectives linked to any auxiliary (e.g., socioeconomic) data.

### 3.3.3. Gridded Approach

The single SDG 11.6.2 Indicator value for Hamburg (in 2016), as calculated with the gridded approach (Section 2.3.2), is 12.33 μg/m$^3$. This is a further 4.2% increase compared to the neighbourhood approach, demonstrating a smoothing effect accompanying the consecutive aggregation, as the higher the spatial resolution, the higher the value of the SDG 11.6.2 Indicator. As shown in 3.2, such a difference (in the order of 7% overall between the gridded and the district approaches) can prove critical for the achievement or not of specific targets.

In addition to the calculated 11.6.2 Indicator, which fully exploits the resolution of the simulated pollutant concentrations, the gridded approach allows for pixel-by-pixel

analysis. Similarly to Figures 9 and 10 shows an easy way to identify hotspots, combining the high levels of pollutant concentrations with high population density.

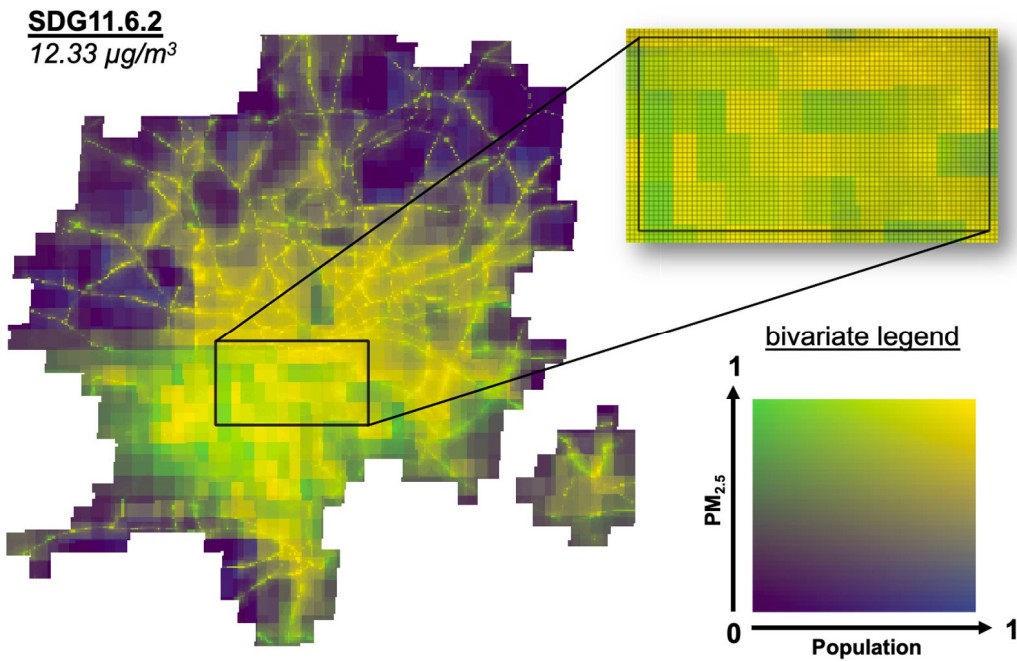

**Figure 10.** Bivariate map that combines population from GHS POP and simulated $PM_{2.5}$ concentrations (normalized values) to identify hotspots of population exposure to particulate matter pollution (yellow) or highly polluted but not populated areas (green), in comparison to low polluted areas, with low (purple) or high (blue) population density.

It becomes clear that the hotspots of population exposure for Hamburg are not only the city center but also the areas near roads. In the industrial and port area of Hamburg, there are high $PM_{2.5}$ concentrations simulated, but the population that is impacted is lower, raising again the need for measures that address both emissions at the source level and exposure at the neighbourhood level.

## 4. Discussion and Policy Implications

### 4.1. Divergent Results with Different Approaches and City Definitions

Despite the fact that SDG 11.6.2 is a Tier 1 indicator, as was shown in the Section 3, the values differ greatly between approaches in some countries and may even affect infringement regarding limit values. While this work does not wish to supplant regulatory and well-established workflows, it is considered important that the stakeholder acquires a firm grasp of the factors driving the national values and understand where uncertainties may be introduced. The UI/UX of the SMURBS SDG 11.6.2 platform had this requirement embedded in its architecture which, in turn, also addressed clearly identified gaps and requirements by a host of city stakeholders of various types [54].

A primary policy implication relevant to the platform stems from the selected city definition and its driving force towards the national indicator value. In particular, the UC value is usually larger than the FUA-based calculation (Section 3.1). The reasoning behind this systematic departure is that the UC hosts the hotspots of population exposure to air pollution, while the surrounding commuting zones included in the FUA are less populated and/or polluted. In cases where the differences between the two calculations are larger, then the geographical patterns of the city show asymmetry depending on the selected approach. From the comparative analysis carried out in Section 3.1 (Table 1) and given that, on average, the country-level SDG 11.6.2 values from SMURBS, UN, and Eurostat approaches maintain a general pattern of FUA < UC < UN < Eurostat, it makes sense to

attempt to quantify the extent to which the UC values can stand as an alternative for the UN SDG 11.6.2 value. Table 1 includes information helpful in framing our discussion on the criticality of the selection of data source and methodology for the calculation of the indicator with respect to policy implications.

As the platform allows for simultaneous provision of both city- and country-level SDG monitoring, it reveals country values driven by a few large urbanities (e.g., Italy, Greece) as well as those that are instead influenced by a multitude of smaller cities (e.g., Germany, the Netherlands). Beyond gaining an understanding of the urban makeup of and resulting influence on country performance, heterogeneity among the national and city values allows for evidence-based awareness of the need for targeted urban policy actions. Part of this conversation about the urban composition should be focused on the applicability and usefulness of certain city definitions that have traditionally been used, and this is where the FUA dilution of SDG 11.6.2 provides a prime example. This is further justification for the ongoing international effort rewiring workflows to base city definitions on an agreed-upon amalgamation of urban density metrics that are relevant for such types of monitoring, such as the degree of urbanisation, as opposed to overestimating the population included in city-level measurements, as in both the urban agglomeration or FUA concept.

While the SMURBS platform utilizes alternate EO data sources as described in Section 2.2.1, the values of the platform tend to be in general agreement with the UN and Eurostat ones (though UC < UN < Eurostat, according to the average of all these values for countries, this tendency is based on some large discrepancies, as there is an almost balanced distribution of countries where one or another indicator appears larger). This consistency enables the CAMS-driven SDG values to serve as a trusted complement to official methods, especially for countries with limited monitoring resources. Modelled information (infused with satellite information and validated by in situ data) provides not only more universal monitoring of indicators like SDG 11.6.2, but also allows for spatial representativity, contributing to a more holistic and continuous picture for policy makers to use as inputs to locate hotspots, focus interventions, and promote action aimed at achieving targets.

Nevertheless, differences between all approaches should always be critically explored to determine whether discrepancies can be attributed to city definition or issues related to inconsistencies or accuracy resulting from the model used, especially in situations where the values for SDG 11.6.2 lie on different sides of limits or guidelines.

Having discussed the shift from national to city reporting, the next natural step is an intra-urban approach, for which the application of an urban-scale air quality model seems to be essential as air quality fields are not available in high resolutions, i.e., lower than 1 km. In this case, the availability of necessary data in different stages of the application remains a critical implementation challenge, mostly technical but also political (see trusted emission inventories):

- Emission inventories, meteorological data, and pollutant boundary conditions: this information, which is necessary for chemistry-transport models, is publicly available for the whole of Europe and many other regions in the world and can be accessed for different years. Since emission inventories are mostly driven by information provided by countries, in regions where there is lack of adequate information or there is an indication of deliberate underestimation of national emissions, then this propagates as an error in the estimation of the indicator values and might be reflected as increased discrepancy between the different available methods.

- In situ air quality data: this is necessary to validate and evaluate a model's output and, in particular, to ensure good performance, improve or correct the model results, apply offsets, or refine emission inventories. Moreover, the comparison of predictions against measurements helps reveal sources, seasonality, or any type of inconsistencies which should be taken into account in the evaluation of the SDG 11.6.2 Indicator.

- Geospatial information for the extent of the appropriate authority unit: the administrative-oriented approach relies on available spatial definitions (and respective data) for districts, neighbourhoods, or other statistical units, which can be accessed via national, city, or municipality sources, usually via open data platforms.

- Population density: the respective population per unit needs to be known for the calculation of the indicator. In addition, one must not only determine the population, but match it with the corresponding authority unit, which may not always be the same all around the world. Cities that lack such information may overcome this barrier through the gridded approach, i.e., by combining urban-scale air quality fields with publicly available UN datasets. The 2021 EU-wide population and housing census has presented a major innovation along the above lines, as key census elements will be delivered on a 1 km square grid, allowing for more flexible exploitation, and facilitating tailored research and accomplishment of policy needs (such as the SDGs' frame).

In the study at hand, to evaluate the consistency of the intra-urban approaches, on top of their unambiguous usefulness for policy making, we compared the UC and FUA concentration values against existing measurements in Hamburg for 2016. In particular, the global WHO's Ambient Urban Air Pollution database [55], which is based on measurements in urban areas, was utilized to retrieve the value of the SDG 11.6.2 Indicator for Hamburg. In this database, the annual concentration average of pollutants is based on a few measurement sites. At times, the calculation is based on only one station representing an entire urban area, or on sites that are not representative and flagged as either only urban background or road sites. Thus, depending on the stations' location, there is a high likelihood of introducing positive or negative bias to the annually averaged concentrations. For the case of Hamburg, the WHO database value is based on four stations—two traffic, one urban, and one site at the airport.

The $PM_{2.5}$ value in the WHO database for Hamburg in 2016 is 14.01 $\mu g/m^3$. This is:

- 18% higher, compared to the district-approach value (11.51 $\mu g/m^3$);
- 15% higher, compared to the neighbourhood-approach value (11.83 $\mu g/m^3$);
- 12% higher, compared to the gridded-approach value (12.33 $\mu g/m^3$).

Thus, by increasing the spatial resolution of population data and $PM_{2.5}$ concentration, a convergence towards the WHO database value is observed. Given the 4 measurement stations taken into account for the extraction of the WHO value, compared to the multiple thousands of grid cells (i.e., 52,568 cells of $100 \times 100$ m$^2$ resolution) derived through the gridded approach, it is obvious that there is a different coverage and potentially representativity of the urban domain.

In the urban-scale approaches, there are areas taken into account which cannot necessarily be considered urban areas but still resemble either municipal or urban areas according to the GHSL UCDB definition. Especially for the intra-urban domain definition, based on the municipal boundaries, there are many such areas taken into account on the outskirts of the city, which are less populated and without either measurements or high air pollution levels (according to the model's simulation), that would lower the value of a single SDG 11.6.2 Indicator value for the entire city. The above, combined with a slight, still acceptable, underestimation of $PM_{2.5}$ by the model (see Section 3.2), are factors that can introduce a certain bias, negative or positive, and should be accounted for before calculating an indicator value via any approach. Nevertheless, the goal of the presented evaluation exercise is not to reproduce the WHO database values, but to improve the SDG 11.6.2 methodologies towards higher spatial representability in urban areas.

It is noteworthy that, apart from the sensitivity to the city extent, population data conventionally refers to residential addresses with no temporal variation. This assumption is considered obsolete, but still the most popular for population-weighted concentration calculations and realistic exposure studies. There exist only a few approaches to incorporate citizens' daily activity into population exposure calculations (e.g., [44,56–58]). Such studies that take population dynamics into account have shown that, particularly in urban areas with a lot of road traffic, the exposure of the population to both $PM_{2.5}$ and $NO_2$ is underestimated. The policy implication here is not aimed at undermining the general applicability of SDG 11.6.2, which is, after all, a proxy for population exposure, but to elicit further caution.

*4.2. Disaggregating SDG 11.6.2 and Enabling Local Action*

One could argue that Eurostat or UN official values and the ensuing databases/dashboards would suffice and that there remains no need for further efforts, as, after all, 11.6.2 is a Tier 1 indicator. The sensitivity of the indicator discussed above, at least, underpins the need for delineating the drivers of the indicator at the national level for all policy matters. The SMURBS platform and intra-urban approach, firstly, aimed to provide such a delineation presented in an eloquent manner. However, as described in Section 1, new policy frameworks, such as the Decade of Action or the Leave No One Behind principle of the SDG framework, create enhanced needs for reporting compared to the ambitious, albeit minimalistic, SDG approach with its global applicability.

Here, the mandate inscribed in the official definition of the SDG indicators themselves, which states they should be "disaggregated, where relevant, by income, sex, age, race, ethnicity, migratory status, disability and geographic location", is realized through the use of EO, whose potential to do so is put to the test [59]. National-level values are disaggregated to city-level values, and cities that drive the indicator are readily identified in the platform, creating a clear entry point for policy interventions. While SDG 11.6.2 is intended to offer a cohesive and comparable global view of the overall air quality between countries, the platform approach and, especially, the disaggregation in the policy relevant groupings of Table 1, demonstrate that this is not always the case. Countries with severe discrepancies between the different sources/approaches for the indicator should evaluate jointly the results of the approaches to, at least, comprehend the cause of uncertainty or evaluate their monitoring network (e.g., representativity, gaps) that feeds the statistical model that ultimately produces the official UN Indicator [60].

The next entry point is the city itself through the intra-urban approaches where the localization of SDGs is realized to the furthest level. This analysis, among others, enables the concurrent view on air quality conditions and population (Figures 9 and 10), echoing the need of a nexus approach. It also enables the identification of environmental inequity issues, as the base layer provided by the approach facilitates integration with more traditional socioeconomic data that are routinely reported at the administrative level. This integration is integral in the discussion of so-called trade-offs and co-benefits of the SDG framework. A city, because of its inherent complexity, cannot afford to address the SDGs' indicators or underlying causes separately, as action towards one indicator may negatively affect other indicators. Such a trade-off can be seen in residential wood burning serving Goal 7 (Affordable and Clean Energy), but simultaneously having a negative impact on SDG 11.6.2. Conversely, climate action (SDG 13) such as improving energy efficiency in buildings will lead to less central heating emissions, thus, an improvement in SDG 11.6.2, or a co-benefit. These interlinkages necessitate spatial disaggregation and modelling capacities, such as the approaches followed in this study, that allow for scenario analysis and quantification of impacts. It should be mentioned that the platform approach, although covering the whole of Europe, also reaches the city level and is fully compatible with the localization of the SDG effort as the discussion on the SDSN SDG Index cities proves.

Both intra-urban approaches are valuable tools for regional or national authorities to identify priority areas for policy action and measures to improve air quality. The organizational levels of an urban area will dictate which kind of information provided by the intra-urban approaches might be of particular interest. For example, in the city of Hamburg, specific measures to reduce air pollution are initiated and controlled by districts and, at times, neighbourhoods. Thus, for this level of policy action, it is necessary to analyse SDG 11.6.2 with the administratively oriented approach. However, when it comes to planning for the road network, the Office for Transport and Mobility Transition is responsible for the entire city. In this case, and for measures that focus on the road network as a major contributor to air pollution, it is necessary to analyse the results of the gridded SDG 11.6.2 approach. In addition to their different applications, both intra-urban approaches enable the rapid identification of hotspots of highly polluted and

populated areas. Bivariate maps (as shown in Section 3.2) are a simple tool that allow easy identification of hotspots on first sight.

### 4.3. Policy and Future Proofing

The approaches discussed in this study are agnostic to limit values or guidelines. As the focus lies on estimating or utilizing PM$_{2.5}$ concentration fields and combining them with the distribution of population, the comparison with any limit values is the last step of the workflow. Moreover, as Figures 5 and 6 demonstrate (which compare Indicator 11.6.2 values for countries and cities, respectively), the process allows a stakeholder to quickly obtain an understanding of how the geographical unit at hand fares with respect to the limit and against other similar units, such as cities with comparable population or geographical/sectoral characteristics.

As noted in Section 1, the Zero Pollution Action Plan target is expected to be met if the rate (33% for the 2005–2019 period) of reducing premature deaths is to be maintained. From the authors' viewpoint, this implies that maintaining and improving the geographic scale of action is key and, ultimately, planning and impacts on air pollution is city-specific, as suggested by the European Commission's Joint Research Centre in their Urban PM$_{2.5}$ Atlas [23]. Both the general applicability to limit values and the detailed data required for city planning are exemplified in the gridded approach, as demonstrated for the city of Hamburg. The comparison of the gridded values requires a little more effort but, at the same time, allows for a detailed analysis of the PM$_{2.5}$ concentration distribution of the total population. Figure 11 shows the distribution of the population which is exposed to different PM$_{2.5}$ concentrations. In terms of the gridded single value of SDG 11.6.2 for Hamburg in 2016 (12.33 µg/m$^3$), 8% of the population has been exposed to values above it (red line). Thus, the calculated indicator refers to the 92% of the urban centre's population. All residents of Hamburg were exposed to PM$_{2.5}$ concentrations above the WHO AQG value in 2016 (blue line). The indicator value lies between the WHO AQG Interim Target 4 and 3 (green and yellow lines, respectively), which indicates an exceedance by a factor of 2–3 of the WHO AQG value; 41% of the population was exposed to PM$_{2.5}$ values above Interim Target 4 (and 2.5% of the population above Target 3). Lastly, according to the less stringent annual limit value of the European Commission, all citizens (but for 1%) were exposed to "safe" levels of PM$_{2.5}$ (black bar).

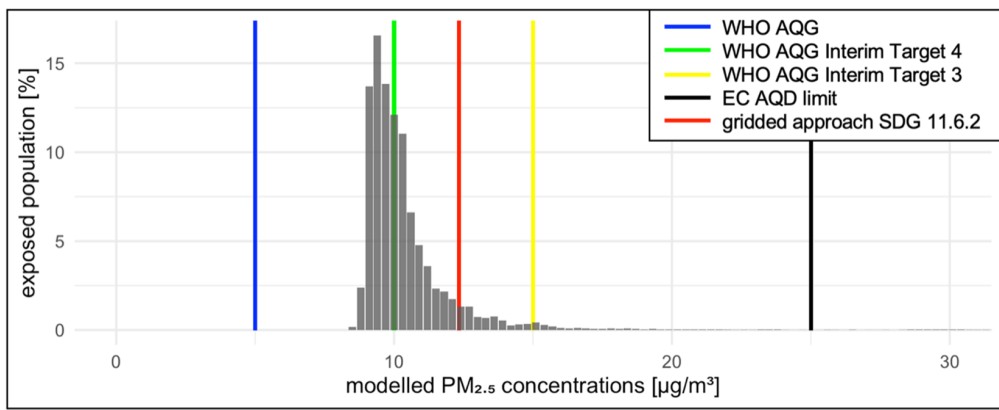

**Figure 11.** Percentage of population in Hamburg, 2019, that is exposed to different PM$_{2.5}$ levels (modelled). The coloured lines show different air quality limit and guideline levels, i.e., WHO AQG (blue), WHO AQG Interim Target 4 (green), WHO AQG Interim Target 3 (yellow), EC limit (black), as well as the SDG 11.6.2 value based on the gridded approach.

This example, as well as the overall intra-urban approach, can be also helpful for epidemiological studies, which are critical in providing concentration response functions that are in turn utilized in estimating the premature deaths of the Zero Pollution Action Plan,

among other limits, guidelines, and plans. Further, improved spatiotemporal information will benefit such studies in the long run.

Future proofing not only refers to policy applicability as discussed above, but also to scalability, modularity, interoperability, and sustainability. The platform has already been implemented for hundreds of cities across Europe. With global products, such as the one coming from the Human Planet Initiative (and Copernicus Emergency Management Service), i.e., the Global Human Settlement suite of products, the only other data requirement is, of course, gridded information of $PM_{2.5}$ so that it can be geographically scaled up. The approach can easily accommodate any such information; in addition, this information is expected to be more readily available and at finer spatial resolution in the years to come. This modularity is accompanied with interoperability with other data products that are expected to arrive over the following period. As already mentioned, the 2021 census across Europe is also expected to be distributed on a 1 km grid. This is by design, and it will enable easier integration with other geospatial data, such as the ones discussed here. In terms of application of these approaches to regions beyond Europe, both current and future data products and services do and will allow for mimicking of such an exercise in other regions. However, as was clear in our results, the urban and intra-urban complexities that influence SDG 11.6.2 need to be taken into account (i.e., geography, city definition, air pollution trends, transport, etc.). The discussion of trade-offs and co-benefits, as well as estimations of vulnerability, risk, and resilience, will surely benefit from such integration. The timeliness is ideal as the Open Data Directive places EO, geospatial, and statistics data on a par, designating them as High Value Datasets.

Lastly, both intra-urban approaches aim at overcoming the bane of SDG localization efforts (data availability, quality, and comparability) and, perhaps more importantly, at aligning with stakeholders' operational needs. The almost exclusive reliance on Copernicus services (i.e., CAMS, CEMS, CLMS, C3S) exemplifies their uptake towards serving the SDGs and guarantees the sustainability of the proposed solutions. Future work enabled by EO data, fueled by remote sensing, investigating and integrating temporal variability into the calculation of indicators, may clarify the blurred picture that annual means can provide, further elucidating a more impactful policy approach.

## 5. Conclusions

The New Urban Agenda highlights that local governments are the prime movers of sustainable urbanization, and the increasing integration of Sustainable Development Goals (SDGs) into policies/action plans serves as an excellent opportunity to help cities engage with more local stakeholders and engage in more inclusive, interdisciplinary, and, therefore, effective decision making.

This work unfolds the great opportunities arising from the above trend through a specific example of an SDG Indicator (i.e., 11.6.2) and a globally recognized urban issue with huge implications on human health and climate, that of air quality. The proposed approaches not only support the monitoring of the SDG Indicator via alternative means, but also attempt to tackle part of the actual problem related to the SDG 11.6.2 objective, to reduce city concentrations of particulate matter and, subsequently, the exposure of citizens to harmful substances. Current official reporting systems (i.e., United Nations, Eurostat), which are mainly based on in situ monitoring networks, can suffer from lack of representativity for the actual diversity of urban conditions, such as not covering smaller cities—even those that have experienced or identified air quality problems—or being vulnerable to policy conflicts with respect to emissions reporting. For this, it is imperative to supplement official reporting with alternative sources of information, such as Earth observation, to enable calculating more representative and holistic values for the indicator based on population density as opposed to differing city definitions.

The presented approaches initiate an application which utilizes objective and non-arbitrary city definitions, as well as air quality (AQ) information derived by Earth observation (EO) platforms (in situ, model, satellite) to deliver national and disaggregated city-level

information on the indicator, delivering an alternative for verifying official reporting and enabling decision making based on evidence at the appropriate level of authorisation. Our analyses revealed that, although general agreement occurred for several countries, comparisons between the indicator approaches yielded differences with respect to European Union and World Health Organization (WHO) limit exceedances, and wide divergences with ensuing exposure impacts. In these cases, it is necessary for countries to realize explicit investigation to delineate the factors that control divergences and for cities to comprehend the nature of differences across their extent. All these disparities carry policy implications, both at the country and city level, suggesting that a critical review of the current monitoring of the SDG 11.6.2 workflow is, perhaps, called for, especially to ensure that it integrates this harmonised city definition and, therefore, remains impactful globally and accomplishes the ultimate ambition.

Such combined workflows should not be limited by the availability of in situ data since modelled and satellite derived data are available, arriving, and improving in a replicable manner. Due to the goal of moving from the European level to the global level, complementing and helping in the efforts of the WHO—as the custodian agency for the indicator—and indicating and communicating the sustainable development agenda with Eurostat on the European level are urgent requirements.

Earth observation and statistics are converging at the research level (e.g., see Big Data for European Statistics, BDES), but are not there yet. The SDG framework provides the common field to experiment and ultimately jump into the "serious" domain of official statistics, while the granularity offered opens untapped opportunities for combining with other statistics, such as health and emissions accounting. Moreover, efforts to improve AQ enhance climate action, and vice versa, as climate change mitigation efforts can improve AQ and, therefore, health and well-being in cities. The COVID era has proven how immediate the improvement in AQ can be, but also how rapidly the situation can be reversed or worsened. Thus, the momentum gained from living with lower AQ should be maintained, which can be helped by more efficient communication of the related co-benefits to climate action. Further bolstering the momentum lies in the environmental justice and equity role of helping to identify disproportionate exposure to air pollution.

EO, as one of the high-value datasets, leads the way towards a bright and shiny future, where consistent, free-of-charge, easy-to-use data are available worldwide and are used to optimize the monitoring of indicators such as SDG 11.6.2 and ultimately improve the situation in air quality, pushing forward progress on the new WHO Air Quality Guideline values.

**Author Contributions:** Conceptualization, J.B., M.O.P.R., E.A., M.K., O.S. and E.G.; methodology, J.B., M.O.P.R., E.A., M.K., O.S. and E.G.; validation, M.O.P.R.; formal analysis, J.B. and M.O.P.R.; investigation, J.B., M.O.P.R. and O.S.; data curation, J.B., M.O.P.R. and O.S.; writing—original draft preparation, J.B., M.O.P.R., E.A., M.K. and O.S.; writing—review and editing, J.B., M.O.P.R., O.S. and E.G.; visualization, J.B. and M.O.P.R.; project administration, E.A. and E.G.; funding acquisition, E.G. (project coordinator). All authors have read and agreed to the published version of the manuscript.

**Funding:** This work was funded by ERA-PLANET (www.era-planet.eu), trans-national project SMURBS (www.smurbs.eu) (Grant Agreement No. 689443) under the EU Horizon 2020 Framework Programme.

**Data Availability Statement:** The data on intra-urban $PM_{2.5}$ concentrations and SDG 11.6.2 indicators for Hamburg in 2016 presented in this study are available on request from the corresponding author. The data from the SMURBS SDG Indicator 11.6.2 Earth Observation Platform are available online through the platform itself and can also be requested in bulk from the corresponding author.

**Acknowledgments:** The authors would like to thank Copernicus Services for the public distribution of Urban Atlas, Corine Land Cover, CAMS regional ensemble reanalysis product, and the suite of products under the Global Human Settlement Layer. We further acknowledge ECMWF for ERA 5 synoptic reanalysis. On behalf of NOA, the co-authors would also like to acknowledge support from the e-shape—EuroGEO Showcases: Applications Powered by Europe project, funded by the European Union's Horizon 2020 research and innovation programme under grant agreement 820852.

The authors acknowledge the valuable contribution of Dimitrios Vallianatos in terms of IT design and implementation of the SMURBS SDG Indicator 11.6.2 Earth Observation Platform.

**Conflicts of Interest:** The authors declare no conflict of interest. The funders had no role in the design of the study; in the collection, analyses, or interpretation of data; in the writing of the manuscript; or in the decision to publish the results.

## Appendix A.

*Appendix A.1. Geographical Map of Studied Cities*

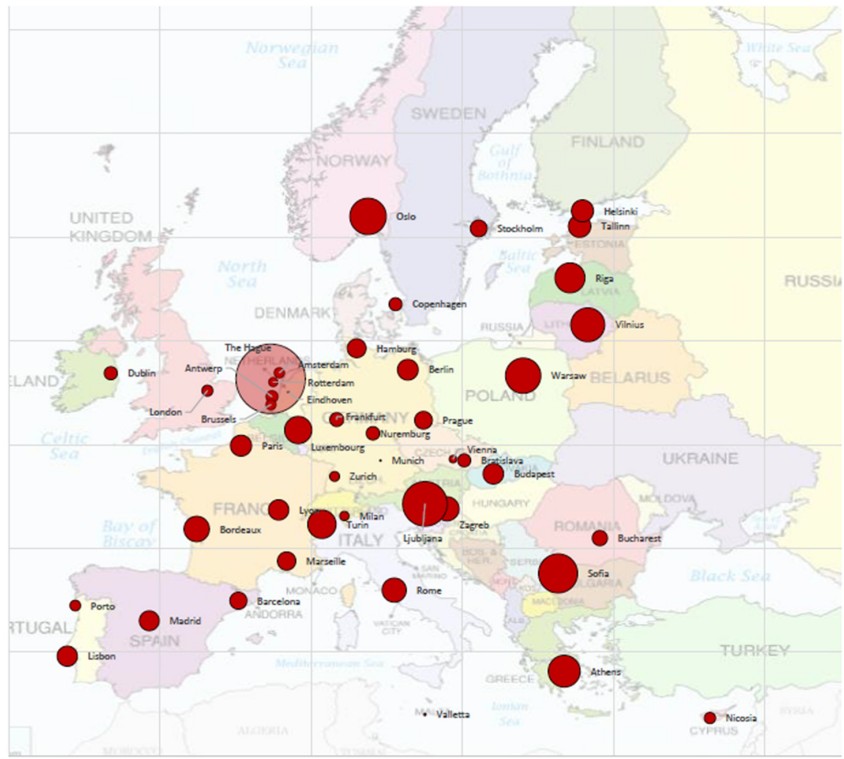

**Figure A1.** A geographical map with the studied cities. Circle sizes correspond to the absolute differences of the annual (2016) PM$_{2.5}$ concentration values (μg/m$^3$) between calculations for the urban centre and the functional urban area (UC-FUA).

*Appendix A.2. Overview of EPISODE-CityChem Setup*

**Table A1.** Overview of EPISODE-CityChem setup for Hamburg (2016).

| CTM Setup with EPISODE-CityChem | Setup for Hamburg 2016 |
|---|---|
| Horizontal domain size (x × y) | 30 × 30 km$^2$ (30,000 × 30,000 m$^2$) |
| Horizontal domain resolution | 100 m |
| Model grid coordinate system | WGS1984 Universal Transverse Mercator (UTM) Zone 32N |
| Vertical dimension | Lowest layer height 17.5 m, 16 vertical layers below 1000 m, vertical top height 3750 m |
| Boundary Conditions | Hourly Copernicus Atmospheric Monitoring Services (CAMS) regional ensemble concentrations |
| Meteorology | Hourly meteorological fields simulated with The Air Pollution Model (TAPM), 1000 m horizontal grid resolution |
| Point source emissions * | 750 sources (federal emission reports, 11.BimSchV) |
| Lin source emissions * | 12,625 road links (CAMS-REG-AP v3.1, OSM) |
| Area source emissions * | 6430 sources, grid resolution 1000 m (CAMS-REG-AP v3.1) |

\* Detailed descriptions of the emission inventories can be found in Ramacher et al., 2020 [44].

*Appendix A.3. Evaluation of CityChem Simulations*

The evaluation of statistical values for daily PM$_{2.5}$ concentrations exhibits model performance with an average Pearson correlation coefficient of r = 0.5. The model tends to underestimate PM$_{2.5}$ concentrations at urban (13ST) and urban background (20VE, 61WB) stations with an annual normalized mean bias (NMB) of −23%, −12%, and −9%, respectively, and a mean bias (MB) up to 3 μg/m$^3$. At the 2 road sites (64KS, 68HB), the model underestimates the measured concentrations with an annual NMB of −17% and −14%. The general underestimation of modelled concentrations is probably due to underestimated emissions and/or boundary concentrations. The agreement between the 2 datasets (predictions and observations) is satisfactory, with their index of agreement (IOA) found to be approximately 60% in all cases but one. All stations for measured vs. modelled PM$_{2.5}$ values show FAC2 values of 0.78–0.88 for daily PM$_{2.5}$ (Table A2), which satisfies the FAIRMODE acceptance criteria of FAC2 > 0.3 for urban dispersion model evaluation [51].

**Table A2.** Statistical evaluation of n pairs of measured and modelled daily PM$_{2.5}$ concentrations at available measurement sites in Hamburg (2016). The statistical measures used are the fraction of modelled values within a factor of two of the observed (FAC2), the mean bias (MB), the normalized mean bias (NMB), the root-mean-square error (RMSE), the correlation (r), and the index of agreement (IOA). The equations for each parameter can be found in Ramacher et al., 2021 [45].

| Site | n | FAC2 | MB | NMB | RMSE | r | IOA |
|---|---|---|---|---|---|---|---|
| 13ST (urban) | 352 | 0.81 | −3.03 | −0.23 | 8.39 | 0.51 | 0.60 |
| 20VE (urban background) | 364 | 0.81 | −1.62 | −0.12 | 7.67 | 0.48 | 0.61 |
| 61WB (urban background) | 364 | 0.78 | −1.18 | −0.09 | 9.26 | 0.29 | 0.53 |
| 64KS (road site) | 347 | 0.82 | −2.49 | −0.17 | 7.91 | 0.51 | 0.60 |
| 68HB (road site) | 363 | 0.88 | −2.27 | −0.14 | 8.29 | 0.52 | 0.62 |

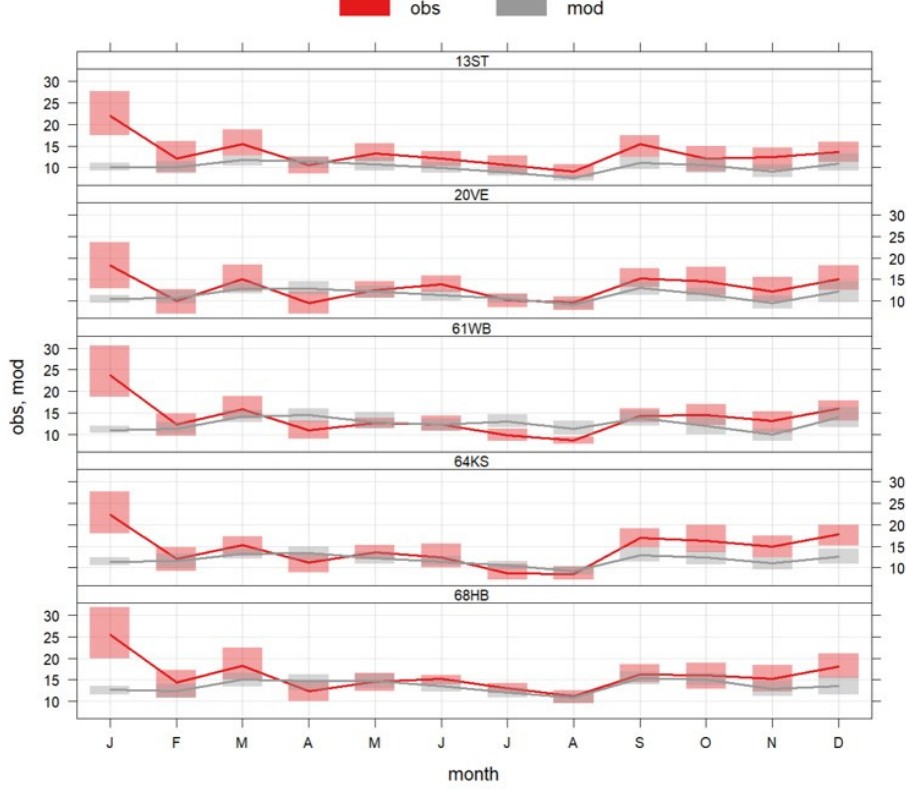

**Figure A2.** Modelled (gray) vs. measured (red) monthly PM$_{2.5}$ concentrations at available measurement sites in Hamburg (2016).

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
