# Peer review of "Localizing SDG 11.6.2 via Earth Observation, Modelling Applications, and Harmonised City Definitions: Policy Implications on Addressing Air Pollution"

_remotesensing, doi:10.3390/rs15041082_

Round 1

Reviewer 1 Report

The manuscript presents an interesting approach towards the impact of scaling in the monitoring of air pollution and the implementation of SDG11 .6.2 via EO means. However, RS data are not used in the manuscript in order to achieve the aim of the paper, but instead data derived from existing platforms are utilized by the authors for which no sufficient information are provided concerning the datasets that these platforms employ in order to compile the data layers or maps. Thus, the connection between RS data and the products of the aforementioned platforms is ambiguous. The manuscript needs to be revised by the authors in order to remove instances of informal speech and the structure especially in the Methodology section should be clearer. The section of Methodology needs to be re-written and the authors must provide a clearer framework of the steps that were followed and the data that were used in each step. A workflow chart would be helpful. Furthermore, references in existing datasets and methodologies that are not part of the methodology that is employed by the author should be reserved for the chapter of Discussion. Finally, the authors should provide more details regarding the input data that is used from the platforms that are mentioned in the manuscript and the input data that was employed in simulation softwares (e.g. EPISODE-CityChem).

The section of Results needs to be revised by the authors in order not to include elements that ought to be incorporated into the Discussion section.

Suggestions

Line 107: Please provide the full name: Key Performance Indicators, since it is the first time mentioned in the manuscript.

Lines 164-170: Please move this paragraph to the Methodology section.

Lines 215-218: Please rephrase in order to provide a clearer meaning for this sentence.

Line 225: Please state clearly which definition of the city is used in this study and provide arguments for your selection.

Line 238: Please change “were not yet published” with “have not been published yet”.

Line 242: Please mention how you incorporate this tool into your methodology framework.

Lines 247-249: Please rephrase in order to provide a clearer meaning

Line 265: Perhaps it would be more useful if this sub-section (2.2.2) could be incorporated into sub-section 2.2.1. In such a case the title of sub-chapter 2.2.1 should change into “Copernicus data processing”.

Line 271: Please change “attained an” into “were incorporated into an”

Line 274: Please provide more details on the “basic statistics” that were used.

Line 275: Please be more specific into what the algorithm calculates.

Line 284: Please change “includes” into “include”

Line 280: The title is suggested to change into “Qualitative comparison with other data sets”

Line 294: Please provide more details on how the qualitative comparison will be achieved.

Line 335: Please justify why you selected Hamburg as a case study.

Line 386: Please justify why the 2μg/m3 threshold was used.

Figure 3: Please change the color of the lines between the EU limit and the WHO Interim Target 2. They look the same. Also there are five lines in the plot (a green and a blue one), which are not explained in the legend.

Lines 414-417: This sentence should be incorporated in the Discussion section.

Line 425-429: Please rephrase the sentence in order to confer a more clear meaning. Is being bordered by the Atlantic Ocean a criterion for low PM2.5 levels? Please explain.

Line 437: Please change “Around” with “Approximately”.

Lines 449-451: This sentence should be incorporated in the Discussion chapter.

Line 467: Please change” do” with “does”.

Line 497: Please change “pulled” with “used”.

Figure 4: Are the lines that stand for the EU limit and WHO Interim Target 2 the same. The lines have the same colour in the legend.

Line 525: Perhaps “city scale” should be replaced with “intra-urban scale”

Line 557: Please change “Analyzing” with “By studying”.

Line 574: Please provide more details regarding the resampling methodology that was followed in the appropriate Methodology sub-section.

Line 664: Please be more specific regarding the EO data sources

Figure 10: Please provide a legend regarding the colored lines of the plot.

Reviewer 2 Report

This manuscript unfolds the great opportunities arising from the above trend through a specific example of an SDG Indicator (i.e., 11.6.2) and a globally recognized urban issue with huge implications on human health and climate, that of air quality. The proposed approaches not only support the monitoring of the SDG Indicator via alternative means, but also attempts to tackle part of the actual problem related to the SDG 11.6.2 objective, to reduce city concentrations of particulate matter and subsequently the exposure of the citizens to harmful substances. This manuscript began by laying out the methodology for calculating SDG 11.6.2 for every city and country in Europe culminating in the SMURBS platform and related comparisons as well as for the calculation using city scale air quality modeling in the case study of Hanburg.

Actually, this manuscript spend large pages to describe the methodology and relative information. However, for me, i have one question, how this study is related with remote sensing ? What is the key objective of your study. Your purpose seems to build up a PM2.5 monitoring platform. Is it suitable to submit as a Article type. Suggest authors to delete some unnecessary description and put the key content. How you do your research? How you combined with remote sensing datasets? and how your research or your work contribute to this field? These questions should be well responsed.

Reviewer 4 Report

In the paper, “Localizing SDG 11.6.2 via Earth observation, modeling applications, and harmonized city definitions: policy implications on addressing air pollution”, the authors investigate and compare two approaches for characterizing the relationship between PM2.5 concentrations and population on a city level.  The authors have completed a thorough study of a very important topic (i.e., PM2.5 air pollution), and include a detailed narrative of the policy implications of the work.  While I have two major concerns, I believe these likely can be addressed fairly quickly and in a straightforward manner, and thus I recommend a minor revision for this paper.  The authors should address the major and minor comments below in their revised manuscript.

Major comments:

1.     Proofreading/rewriting/editing of the text is necessary, as some sentences do not fully make sense and can be broken up/rewritten.  One example occurs in Lines 159-170, but please conduct a thorough proofreading and editing of the entire manuscript.

2.     This paper seems to be heavily weighted towards the analysis of in situ PM2.5 data, population data, and model datasets.  However, this is for the journal Remote Sensing.  Thus, I suggest adding some discussion to place more emphasis on the relationship of this work to remote sensing in both the abstract and introduction.

Minor comments:

1.     Line 156: SMURBS?  Please define/explain.

2.     Line 344: Define FAC2.

3.     Lines 344-345: I suggest discussing other statistics shown in Table A2.

4.     Line 360: I suggest keeping units consistent (i.e., km or m).

5.     Line 590: In Figure 8, it looks like “dark blue” is really “purple”, correct?

6.     Line 811: Figure 10 is not labeled correctly.  This should be Figure 9, correct?  The figure before this one is Figure 8.  Also, the text in Line 798 reads “Figure 11”.  Shouldn’t this be Figure 9?  Please check all figure numbers for consistency with the text.

7.     Lines 810-814 (the current Figure 10): Please edit the “PM25” on the x and y axes to “PM2.5”.  Also, please add units to the x axis label.

8.     Lines 843-898: In the Conclusions section, please redefine all acronyms.

9.     Lines 895-898: This seems like a run-on sentence.  Please break it up into at least two sentences. Also, it should be “data are” not “data is”.  The “data is” occurrence happens in other areas of the text.  Please also fix these.

Round 2

Reviewer 1 Report

The manuscript has been
sufficiently improved to warrant publication in Remote Sensing

Reviewer 2 Report

Authors have addressed my concerns.

Reviewer 3 Report

The authors have incorporated the comments and improved the revised manuscript accordingly. The manuscript can therefore be accepted for publication in its current form.